# Non-canonical Wnt signaling regulates junctional mechanocoupling during angiogenic collective cell migration

Joana R Carvalho, Isabela C Fortunato, Catarina G Fonseca, Anna Pezzarossa, Pedro Barbacena, Maria A Dominguez-Cejudo, Francisca F Vasconcelos, Nuno C Santos, Filomena A Carvalho, Claudio A Franco*

Instituto de Medicina Molecular, Faculdade de Medicina, Universidade de Lisboa, Lisbon, Portugal

**Abstract** Morphogenesis of hierarchical vascular networks depends on the integration of multiple biomechanical signals by endothelial cells, the cells lining the interior of blood vessels. Expansion of vascular networks arises through sprouting angiogenesis, a process involving extensive cell rearrangements and collective cell migration. Yet, the mechanisms controlling angiogenic collective behavior remain poorly understood. Here, we show this collective cell behavior is regulated by non-canonical Wnt signaling. We identify that Wnt5a specifically activates Cdc42 at cell junctions downstream of ROR2 to reinforce coupling between adherens junctions and the actin cytoskeleton. We show that Wnt5a signaling stabilizes vinculin binding to alpha-catenin, and abrogation of vinculin in vivo and in vitro leads to uncoordinated polarity and deficient sprouting angiogenesis in *Mus musculus*. Our findings highlight how non-canonical Wnt signaling coordinates collective cell behavior during vascular morphogenesis by fine-tuning junctional mechanocoupling between endothelial cells.

DOI: https://doi.org/10.7554/eLife.45853.001

*For correspondence:
cfranco@medicina.ulisboa.pt

**Competing interests:** The authors declare that no competing interests exist.

## Introduction

Morphogenesis is driven by coordinated and dynamic cell movements, which are regulated by a combination of chemical and physical cues (*Jaalouk and Lammerding, 2009*). Morphogenic cues are sensed and read at the single cell-level, yet biomechanical information is relayed to and integrated by neighboring cells leading to tissue-level collective cell behaviors. These emergent collective behaviors arise by mechanically coupling cadherin-based adhesion and actomyosin-based contraction, allowing propagation of cell-cell interactions across large cell populations (*Friedl and Mayor, 2017*; *Lecuit and Yap, 2015*; *Yap et al., 2018*). One of such morphogenic processes is the formation of blood vessels. Vascular morphogenesis occurs mainly through sprouting angiogenesis, a process where endothelial tip cells lead the vascular sprout, migrate and invade into avascular tissues in response to pro-angiogenic molecules. Endothelial stalk cells follow tip cells contributing to sprout elongation and branch formation through proliferation and migration (*Potente and Mäkinen, 2017*). Although sprouting angiogenesis is considered a collective cell migration process (*Friedl and Gilmour, 2009*; *Vitorino and Meyer, 2008*), little is known about the mechanisms regulating this collective behavior. Recently, endothelial cell front-rear polarity has emerged as a crucial regulator of collective behavior in sprouting angiogenesis. In fact, endothelial cell (EC)-specific deletion of NCK1/2 and Cdc42 impairs cell polarity, which correlates with decreased sprouting efficiency (*Dubrac et al., 2016*; *Laviña et al., 2018*). However, the mechanisms controlling and coordinating polarity patterns of endothelial cells during sprouting angiogenesis remain elusive.

**eLife digest** When a new blood vessel is created, a leader cell branches out from the lining of an existing vessel before being joined by other cells moving together in the same direction. A protein called Wnt5a regulates this process by helping the cells to orient themselves and finely coordinating their migration, but the exact details of this mechanism are still unclear.

One way that cells can communicate is by touching and physically exerting forces on each other. This is made possible by structures called cellular junctions, which are present at the interface between cells. These can transmit forces within a tissue because they are connected with elements that form the cells' internal skeletons. A protein known as vinculin is involved in these connections.

To find out what role Wnt5a plays in cell migration, Carvalho et al. prevented blood vessel cells from creating the protein. The results showed that Wnt5a helps cells to move together by stabilizing vinculin at cell junctions. This strengthens the physical communication between cells and allows them to efficiently coordinate their movements. Indeed, in the mouse retina, deleting vinculin from cells that make blood cells impaired the formation of new blood vessels.

Problems in the way that blood vessels grow are very common in the human population. In addition, Wnt5a is linked to cancer progression, which also relies on coordinated movement of cells. A better grasp of the role of this protein could therefore be relevant to understand how blood vessels are formed, but also how certain cancers invade surrounding tissues.

DOI: https://doi.org/10.7554/eLife.45853.002

Recent reports showed that non-canonical Wnt signaling, a known regulator of cell migration and cell polarity in key morphogenic events such as gastrulation, neural tube closure, fur orientation, and ureteric bud formation (*Gray et al., 2011*; *Yang and Mlodzik, 2015*), also controls sprouting angiogenesis and vascular remodeling (*Franco et al., 2016*; *Korn et al., 2014*). Non-canonical Wnt signaling was shown to control vascular remodeling by blocking excessive vessel regression in a flow-dependent manner (*Franco et al., 2016*; *Korn et al., 2014*). In this context, non-canonical Wnt signaling modulates the threshold for flow-dependent EC polarization, inducing premature vessel regression, and leading to a decrease in vessel density (*Franco et al., 2016*). In parallel, abrogation of endothelial non-canonical Wnt ligands also leads to reduce sprouting efficiency (*Franco et al., 2016*; *Korn et al., 2014*). Yet, it remains unresolved how mechanistically non-canonical Wnt signaling regulates sprouting angiogenesis.

Here, we have established a simple assay to measure endothelial collective cell behavior in vivo and in vitro using axial polarity histograms. Using this assay, we uncovered a novel Wnt5a pathway that stabilizes the binding of vinculin to α-catenin at adherens junctions, and consequently the efficient coupling between adherens junctions and the actin cytoskeleton in endothelial cells. We showed that vinculin loss-of-function impairs collective polarity in vivo and in vitro, leading to deficient sprouting angiogenesis. Overall, we propose that non-canonical Wnt signaling coordinates collective cell behavior during vascular morphogenesis by fine-tuning junctional mechanocoupling between endothelial cells.

## Results

### Non-canonical Wnt signaling is required for the coordination of collective cell polarity

Non-canonical Wnt signaling deficiency leads to impaired sprouting angiogenesis, a process that requires extensive cell migration (*Franco et al., 2016*; *Korn et al., 2014*). To investigate the role of non-canonical Wnt ligands in endothelial cell migration, we used a well-characterized model of collective cell migration, the scratch-wound assay (*Tambe et al., 2011*). Wnt5a is the major non-canonical Wnt ligand operating in vivo (*Franco et al., 2016*; *Korn et al., 2014*) and in vitro (*Figure 1—figure supplement 1A*). In the scratch-wound assay, siRNA-mediated knockdown (KD) of Wnt5a, hereafter siWNT5a, significantly impaired wound closure and straightness of cell migration without affecting cell velocity in human umbilical vein endothelial cells (HUVECs) (*Figure 1A*). Accordingly, single cell tracking highlighted coordinated collective behavior in control siRNA (siControl) cells,

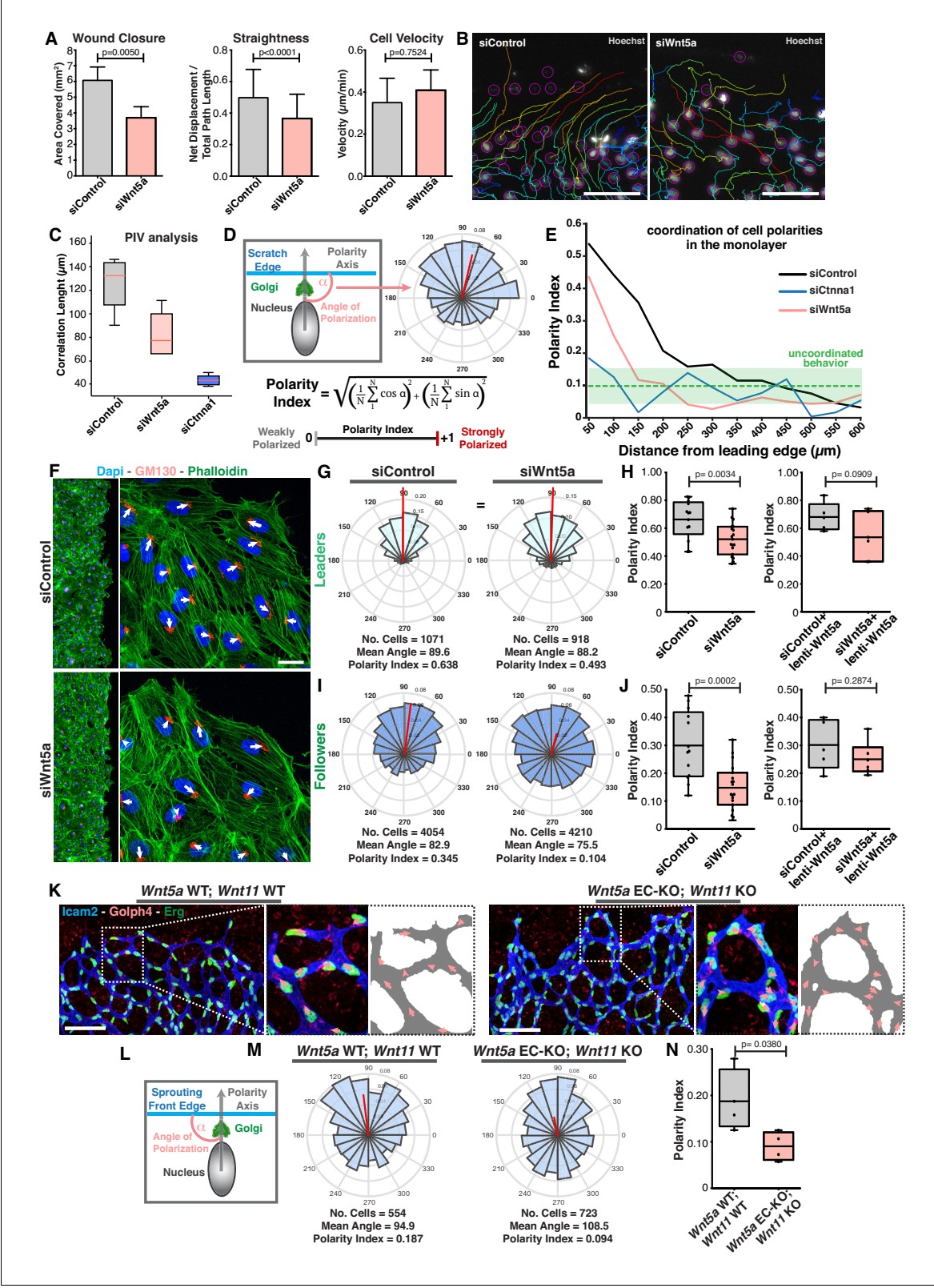

**Figure 1.** Wnt5a regulates endothelial collective cell migration. (**A**) Quantification of wound closure, straightness and cell velocity over the course of 16 hr migration in siControl (n = 100 cells, from two independent experiments) and siWNT5a (n = 100 cells, from two independent experiments) transfected cells. Data are mean ± SEM, p-values from unpaired t-test. (**B**) Wound edge of siControl (left) and siWNT5a (right) transfected cells showing individual cell trajectories within the monolayer. Circles indicate cell nuclei. Scale bar, 50 μm. (**C**) Correlation length box plots from siControl (n = 6),

*Figure 1 continued on next page*

*Figure 1 continued*

siWNT5a (n = 8) and siCtnna1 (n = 3). (D) Polarity axis of each cell was defined as the angle (α) between the scratch edge and the cell polarity axis, defined by the vector drawn from the center of the cell nucleus to the center of the Golgi apparatus. The polarity index was calculated according to the formula and it was used as a measure for collective polarization. (E) Polarity index as function of the distance from the leading edge (μm) in HUVECs monolayers, binning data every 50 μm. Green area corresponds to the mean ± SD of the PI obtained of siCtnna1 cells across the monolayer, excluding leader cells. (F) Representative images of scratch-wound assay showing polarity angles of siControl and siWNT5a KD endothelial cells. Scale bar, 50 μm. (G) Angular histograms showing the distribution of polarization angles of leader cells from siControl (n = 13 images, from six independent experiments) and siWNT5a (n = 19 images, from eight independent experiments). (H) Polarity index box plots of non-infected siControl (n = 13 images, from six independent experiments) and siWNT5a (n = 19 images, from eight independent experiments) leader cells or from siControl (n = 5 images, from three independent experiments) and siWNT5a (n = 6 images, from three independent experiments) leader cells transduced with WNT5a-V5 lentiviruses. p-values from unpaired t-test. (I) Angular histograms showing the distribution of polarization angles of follower cells from siControl (n = 13 images, from six independent experiments) and siWNT5a (n = 19 images, from eight independent experiments). (J) Polarity index box plots of non-infected siControl (n = 13 images, from six independent experiments) and siWNT5a (n = 19 images, from eight independent experiments) follower cells or from siControl (n = 5 images, from three independent experiments) and siWNT5a (n = 6 images, from three independent experiments) follower cells transduced with Wnt5a lentiviruses. p-values from unpaired t-test. (K) Representative images of sprouting fronts from *Wnt5a* WT; *Wnt11* WT and *Wnt5a* EC-KO; *Wnt11* KO mouse retinas labeled for EC nuclei (Erg, green), lumen (Icam2, blue/gray) and Golgi (Golph4, red). Each insert shows corresponding image segmentation of the vascular plexus showing axial polarity vectors (red) and lumen of blood vessels (gray). Scale bar, 200 μm. (L) Polarity axis of each cell was defined as the angle (α) between the sprouting front edge and the cell polarity axis, defined by the vector drawn from the center of the cell nucleus to the center of the Golgi apparatus. (M) Angular histograms showing the distribution of polarization angles of endothelial cells at the vascular sprouting front from *Wnt5a* WT; *Wnt11* WT (n = 4 retinas) and *Wnt5a* EC-KO; *Wnt11* KO (n = 4 retinas) mouse retinas. (N) Polarity index box plots of endothelial cells from *Wnt5a* WT; *Wnt11* WT (n = 4 retinas) and *Wnt5a* EC-KO; *Wnt11* KO (n = 4 retinas) mouse retinas. p-values from unpaired t-test.

DOI: https://doi.org/10.7554/eLife.45853.003

The following figure supplement is available for figure 1:

**Figure supplement 1.** WNT5a, not WNT11, regulates collective behavior in vitro.

DOI: https://doi.org/10.7554/eLife.45853.004

whereas siWNT5a cells showed uncoordinated migration paths (*Figure 1B*). The correlation length calculated from particle image velocimetry (PIV) analysis (*Ng et al., 2012*; *Petitjean et al., 2010*), confirmed loss of coordinated cell migration in siWNT5a cells, although not to the same extent as in cells treated with siRNA against alpha-E-catenin (α-catenin/CTNNA1), a crucial component of adherens junctions and indispensable for collective cell migration (*Figure 1C*) (*Bazellières et al., 2015*).

Axial polarity correlates with the direction of migration in endothelial cells in vivo and in vitro (*Franco et al., 2015*; *Kwon et al., 2016*). Taking advantage of this feature, we generated a simplified method, compared to PIV analysis, to quantify the degree of coordination between cells by measuring the front-rear cell polarity (nucleus-to-Golgi apparatus axis) at the population level. The angular histogram of axial polarities relative to the wound-edge displays the distribution of cell polarities in the monolayer relative to the wound-edge (*Figure 1D*). As a measure of collective polarization, we defined a polarity index (PI, see Materials and methods), which ranges from 1 (strongly polarized) to 0 (random distribution) (*Figure 1D*). The PI represents the length of the mean resultant vector (*Berens, 2009*). Using this approach, we measured PIs in consecutive 50 μm-wide areas from the leading edge towards the monolayer (details in Materials and methods). As expected, siCTNNA1 led to a generalized poor collective coordination of polarities demonstrated by low PIs throughout the monolayer (*Figure 1E*). According to the PI equation, perfect randomization should give a PI = 0. However, α-catenin KD cells shows PI >0, which highlights a polarity bias caused by geometrical constraints that are generated by the free space-cell monolayer interface. Therefore, we used the polarity patterns of siCTNNA1 cells to define the threshold of PI that defines uncoordinated behavior. We established this PI threshold by determining the mean ± SD of the results obtained from the siCTNNA1 experiments across the monolayer. For the calculation of the mean value, we excluded the first row of cells, as these were strongly affected by wound-monolayer asymmetry, leading to a stronger polarity towards the wound. Taking these rules, we defined the PI threshold for uncoordinated migration at PI = 0.14 (corresponding to the upper limit of the mean ± SD, PI = 0.1 ± 0.04, in α-catenin KD experiments (*Figure 1E*). SiControl cells showed coordination of cell polarities up to ~300 μm from the leading edge (*Figure 1E*). Remarkably, siWNT5a cells showed uncoordinated polarity starting at ~150 μm from the leading edge (*Figure 1E*).

In the wound assay, coordinated migration emerges because leader cells, localized at the edge of the monolayer, are polarized due to the presence of a free edge, and instruct follower cells'

directionality of migration through force transmission at adherens junctions (*Etienne-Manneville and Hall, 2001*; *Friedl and Mayor, 2017*). To understand the extent to which the polarization patterns of leaders and followers were affected in their polarization patterns, we measured the PI for leaders (1st row of cells) and followers (2-5th row of cells) separately (*Figure 1F*). Leader cells showed polarization towards the leading edge above random in all three groups: siControl (PI = 0.638), siWNT5a (PI = 0.493) and siCTNNA1 cells (PI = 0.358) (*Figure 1G,H*; *Figure 1—figure supplement 1B,C*). However, siWNT5a (PI = 0.104) and siCTNNA1 (PI = 0.101) follower cells showed randomized polarity patterns whilst siControl follower cells displayed coordinated polarity patterns (PI = 0.345) (*Figure 1I,J*; *Figure 1—figure supplement 1B,C*). Defects in collective polarity in siWNT5a follower cells were rescued by re-expression of exogenous WNT5a (*Figure 1J*).

Cryptic lamellipodia in follower cells have been associated with collective cell migration (*Das et al., 2015*). Thus, we examined if WNT5a plays a role in the formation of these pro-migratory structures. We observed that WNT5a deficiency did not compromise the formation of cryptic lamellipodia but it affected their orientation toward the leading edge (*Figure 1—figure supplement 1D, E*). Taken together, these results indicate that WNT5a signaling is necessary to coordinate the behavior of follower cells at the population-level.

In vivo, endothelial tip cells lead the vascular sprout, whilst endothelial stalk cells follow tip cells and contribute to sprout elongation (*Potente et al., 2011*). In order to evaluate if *Wnt5a* also regulates collective cell polarity in vivo, we calculated PIs for endothelial cells at the vascular sprouting front in control and non-canonical Wnt signaling-deficient mouse retinas (*Figure 1K,L*). Remarkably, we observed a significant decrease in polarity patterns of mutant retinas compared to WT retinas, similar to the effect in the in vitro experiments. We observed collective polarization (PI = 0.187) in control retinas demonstrating that the PI is able to capture collective behavior during sprouting angiogenesis. Whilst, non-canonical Wnt ligand-deficient showed a PI close to randomization (PI = 0.094) (*Figure 1M,N*). Thus, endothelial-derived non-canonical Wnt signaling is required for the coordination of collective cell polarity in vitro and in vivo.

## Non-canonical Wnt signaling regulates mechanical tension at Adherens junctions

To understand how Wnt5a mechanistically controls collective behavior, we analyzed its effects on the adherens junction complex, a key mediator of collective cell migration (*Tambe et al., 2011*). We first characterized the different junctional arrangements in endothelial cells, which are associated with low or high junctional tension (*Huveneers et al., 2012*). We observed that siWNT5a cells had a significant decrease in the frequency of high-force serrated junctions, and a concomitant increase in the frequency of low-force reticular junctions (*Figure 2A,B*). Reduction in the number of high-force junctions correlated with a decrease in the association between VE-cadherin and actin stress fibers (*Figure 2C,D*), suggesting that Wnt5a depletion might negatively impact on force transmission through adherens junctions.

To test this hypothesis, we used atomic force microscopy to probe mechanical strength of cell-cell interactions (*Figure 3A*). Control-control cell interactions required on average 1.0 fJ (1.0 × $10^{-15}$ J) of work (energy) for complete cell-cell detachment (*Figure 3B*). siWNT5a-siWNT5a cell interactions required significant less work (0.5 fJ; p<0.0001) for complete cell separation (*Figure 3B*). EGTA treatment, which chelates extracellular calcium and abolishes cadherin-dependent interactions, significantly reduced the strength of interactions between siControl cells, and canceled the differences between siControl and siWNT5a conditions (*Figure 3B*). VE-cadherin-depleted cells showed a very similar strength of interaction as EGTA-treated cells (*Figure 3B*). A detailed analysis of the frequency of detachment force of each cell-cell interactions in siVE-cadherin condition highlights that the majority of strong cell-cell contacts are mediated by VE-cadherin homophilic interactions (*Figure 3C*). These are significantly reduced in siWNT5a cells (*Figure 3C–F*), suggesting that Wnt5a signaling increases the strength of cell-cell interactions through adherens junctions.

Strength of adhesion at adherens junctions relies on efficient coupling between the cytoplasmic VE-cadherin C-terminus tail and the actin cytoskeleton (*Gumbiner, 2005*). To confirm that WNT5a regulates tension in VE-cadherin, we used previously characterized FRET-based VE-cadherin tension sensors (*Conway et al., 2013*). In Wnt5a-depleted cells, VE-cadherin FRET efficiency was significantly higher than in siControl cells, implying lower level of junctional tension (*Figure 3G,H*). Force-insensitive VE-cadherin FRET sensors showed similar levels between siControl and siWNT5a cells

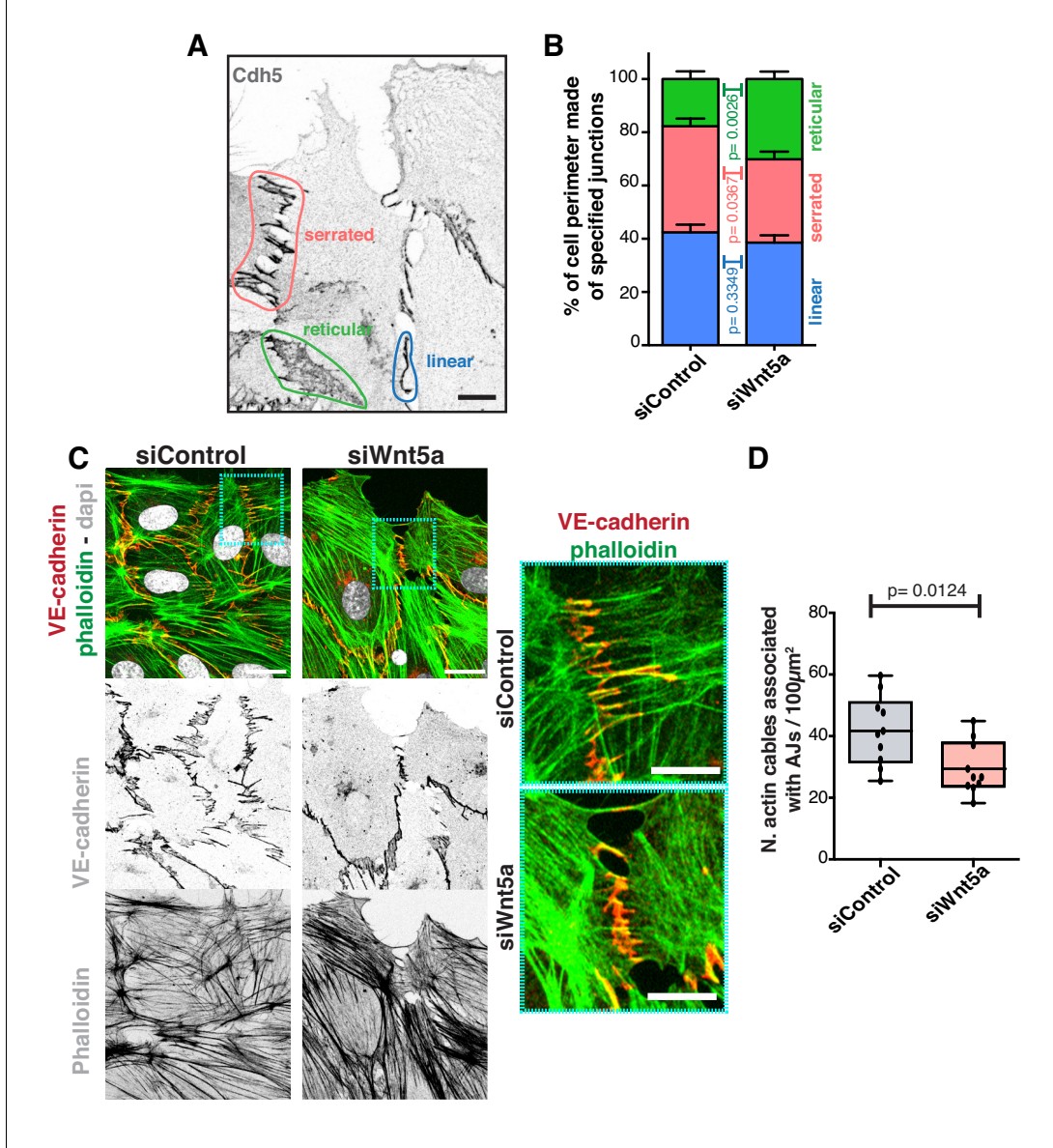

**Figure 2.** Wnt5a regulates adherens junctions' organization. (**A**) Example of the distinct junctions' morphologies in endothelial cells labeled for adherens junctions (VE-Cadherin) showing: linear (blue), serrated (red) and reticular (green). Scale bar, 10 µm. (**B**) Quantification of cell perimeter (%) composed of linear (blue), serrated (red) and reticular (green) in siControl (n = 22 leader and n = 40 follower cells, from four independent experiments) and siWNT5a (n = 40 leader and n = 46 follower cells, from six independent experiments) transfected cells. Data are mean ± SD and p-values from unpaired t-test. (**C**) Detail of wound edge of HUVECs showing the association of actin stress fibers (phalloidin) to the adherens junctions (VE-Cadherin) in siControl and siWNT5a transfected cells. Nucleus labeled with Dapi. Scale bar, 20 µm. Blue squares show a higher magnification of the association of actin filaments (phalloidin) and adherens junctions (VE-Cadherin) in siControl and siWNT5a cells. Scale bar, 10 µm. (**D**) Quantification of the number of actin stress fibers connected to VE-cadherin positive cell-cell junctions in siControl and siWNT5a cells. N = 10 images, from two independent experiments. Data are mean ± SD, and p-values from unpaired t-test.

DOI: https://doi.org/10.7554/eLife.45853.005

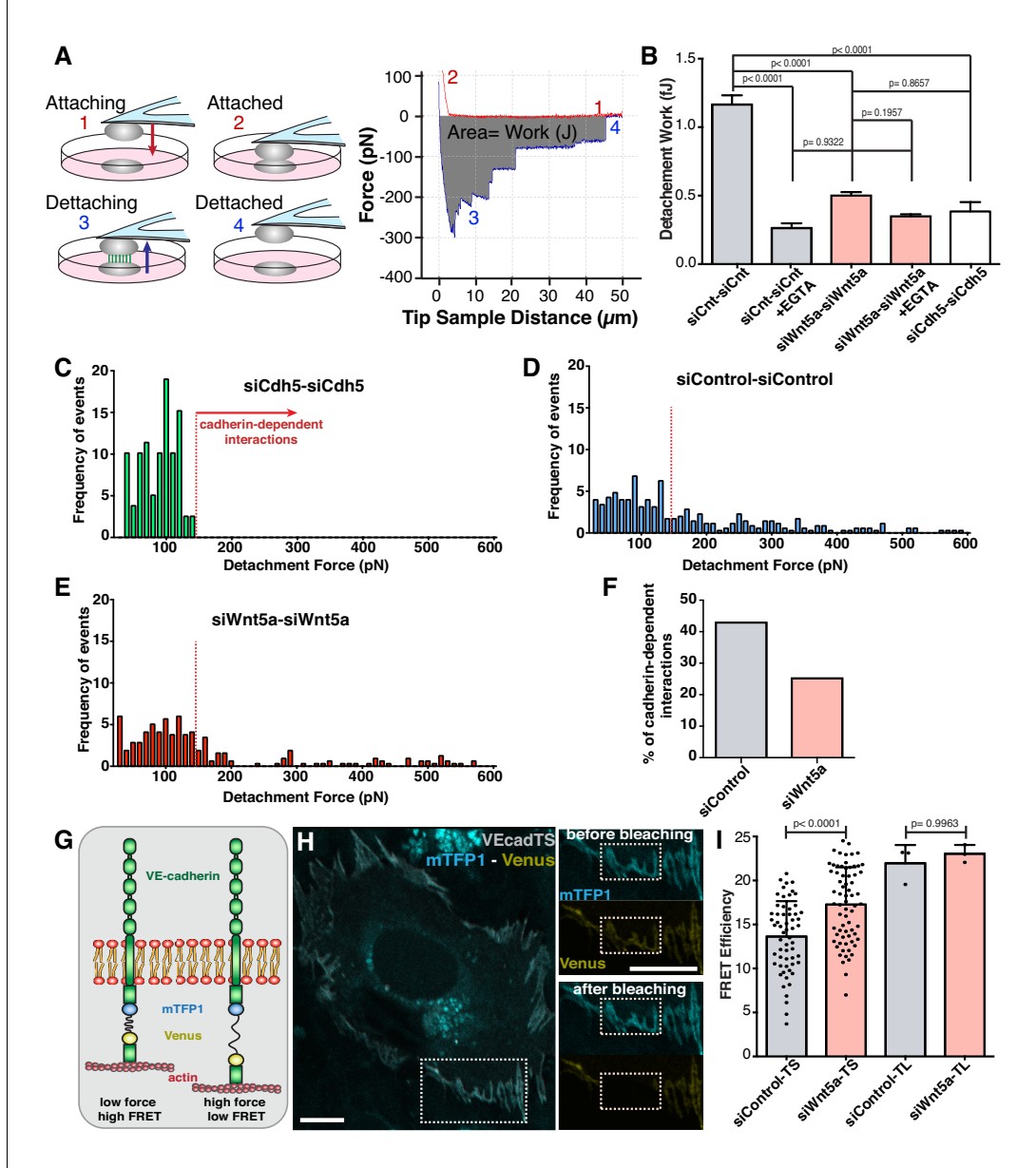

**Figure 3.** Wnt5a signaling strengthens adherens junctions and enhances cell-cell force transmission. (**A**) Diagram depicting the four steps involved in cell-cell adhesion measurements using atomic force microscopy (AFM), as well as its correspondence in the force-distance curves: (1) Attaching – cell attached to the tipless cantilever is lowered to make contact with another cell at the bottom; (2) Attached – cells establish cell-cell contact; (3) Detaching – the upper cell is pulled in order to break the cell-cell contact previously established; (4) Detached – cells are again fully separated. The gray area between the approach (red) and retraction (blue) curves corresponds to the value of work (energy necessary to overcome the cell-cell adhesion). The total force necessary to separate the two cells can also be obtained from the yy axis. (**B**) Quantification of the work necessary for cell-cell detachment in siControl with (n = 155 cell-cell interactions, from five independent experiments) or without EGTA (n = 395 cell-cell interactions, from five independent experiments), siWNT5a with (n = 205 cell-cell interactions, from six independent experiments) or without EGTA (n = 299 cell-cell interactions, from six independent experiments) and siCdh5 (n = 80 cell-cell interactions, from one experiment) transfected cells. Data are mean ± SEM, p-values from multiple comparisons in one-way ANOVA. (**C**) Maximum detachment force histogram for siCdh5 transfected cells (n = 80 cell-cell interactions, from one experiment). Data obtained from one independent experiment. (**D**) Maximum detachment force histogram for siControl transfected cells (n = 395 cell-cell interactions, from five independent experiments) (**E**) Maximum detachment force histogram for siWNT5a transfected cells (n = 299 cell-cell interactions, from six independent experiments). (**F**) The percentage (%) of cadherin-dependent interactions was calculated by dividing the number of events with detachment force above 150pN by the total number of events on each condition. The quantification of the percentage of the Cadherin-dependent interactions was based on the result obtained from the siCdh5-siCdh5 detachment force histogram (in panel C). (**G**) Diagram showing the molecular structure and mechanism of action of the FRET VE-cadherin tension sensor (VE-Cad TS) or VE-cadherin tailless sensor (VE-Cad TL). (**H**) HUVEC expressing VE-Cad TS undergoing FRET acceptor photobleaching at the adherens junction. Squares show the cell

*Figure 3 continued on next page*

*Figure 3 continued*

junction before (top) and after photobleaching (bottom). Scale bar = 10 µm. (I) Quantification of FRET efficiency in siControl (n = 51 cell-cell junctions, from six independent experiments) and siWNT5a (n = 69 cell-cell junctions, from six independent experiments) transfected cells expressing either VE-Cad TS or the tailless biosensor lacking the β-catenin binding-domain, VE-Cad TL (n = 3 cell-cell junctions, from one experiment for both siControl and siWNT5a conditions). Mean ± SD, p-values from unpaired t-test.
DOI: https://doi.org/10.7554/eLife.45853.006

(*Figure 3I*). Taken together, these data demonstrate that Wnt5a signaling promotes high tension at the VE-cadherin intracellular domain and strengthens cell-cell interactions.

## Non-canonical Wnt signaling regulates vinculin stability at adherens junctions to reinforce junctional mechanocoupling

Next, we investigated why loss of siWNT5a results in decreased coupling between adherens junctions and the actin cytoskeleton. First, we quantified the expression levels of key junctional proteins. We confirmed that levels of VE-cadherin, ß-catenin, α-catenin, or vinculin were unaltered between control and Wnt5a-deficient cells (*Figure 4A,B*). Next, we assessed the spatial distribution of components of the VE-cadherin complex by co-localization experiments (*Figure 4C,D*). Interestingly, we observed a significant decrease of VE-cadherin co-localization with vinculin in siWNT5a cells but no change in co-localization with other junctional proteins (*Figure 4D*). We further confirmed a specific decrease in vinculin recruitment to VE-cadherin in siWNT5a cells by proximity ligation assay (PLA) and co-immunoprecipitation (co-IP) in wounded monolayers (*Figure 4E–H*). Altogether, these results indicate that Wnt5a is important to recruit and/or to stabilize vinculin binding to adherens junctions, which in turn is necessary for efficient collective cell polarity.

Vinculin binds adherens junctions via α-catenin. It has also been proposed that a conformational change in α-catenin promotes vinculin recruitment and binding to adherens junctions (*le Duc et al., 2010*; *Yao et al., 2014*; *Yonemura et al., 2010*). To test whether the impaired vinculin co-localization with VE-cadherin arises from defective α-catenin conformational change or from the inability to recruit vinculin once opened, we used a specific antibody that recognizes α-catenin in its open conformation (α18 antibody) (*Yonemura et al., 2010*). α18 antibody-VE-cadherin co-localization showed a significant but mild decrease (~15%) in Wnt5a-depleted cells. Yet, the decrease in vinculin-α18 antibody co-localization was stronger (~32%) in these same cells (*Figure 5A,B*), suggesting a possible defect in vinculin junctional localization even when α-catenin is in its open conformation. To clarify whether WNT5A affects recruitment or stabilization of vinculin to junctions, we quantified the dynamics of vinculin recruitment to adherens junctions at newly formed cell-cell junctions by performing a calcium-switch experiment in siControl and siWNT5a cells. Remarkably, the initial dynamics of vinculin recruitment were similar between siControl and siWNT5a cells. However, a significant decrease of VE-cadherin-vinculin co-localization in siWNT5a cells was observed 30 min after junction reassembly (*Figure 5C*). This suggests that rather than controlling its initial recruitment, Wnt5a signaling regulates vinculin stabilization at junctions.

## Vinculin is necessary for collective cell polarity

The role of vinculin in adherens junctions' mechanical coupling between cells, and in the regulation of collective behavior have been recently established in in vitro studies (*Bazellières et al., 2015*; *Seddiki et al., 2018*). Accordingly, vinculin loss-of-function (LOF) in the scratch-wound assay results in impaired collective cell polarity and migration in vitro, as reflected by the decrease in the closure rate (*Figure 6A–C*). In contrast, the role of vinculin in collective cell migration in vivo remains controversial (*Alatortsev et al., 1997*; *Han et al., 2017*). Thus, we next evaluated the relevance of vinculin in collective polarity in vivo, using the mouse retina model of angiogenesis. We crossed the *Vinculin* floxed mouse (*Zemljic-Harpf et al., 2007*) together with the *Pdgfb*-iCre mouse (*Claxton et al., 2008*) to genetically abrogate vinculin expression in endothelial cells in post-natal mice. *Vinculin* endothelial-specific KO (EC-KO) mice showed decreased radial expansion, decreased vessel density (*Figure 6D,E*), and a significant increase in the number of vessel regression profiles (*Figure 6F*). Strikingly, analysis of polarity patterns of endothelial cells at the sprouting front demonstrated that *Vinculin* EC-KO have a significant decrease in PI when compared with control littermates

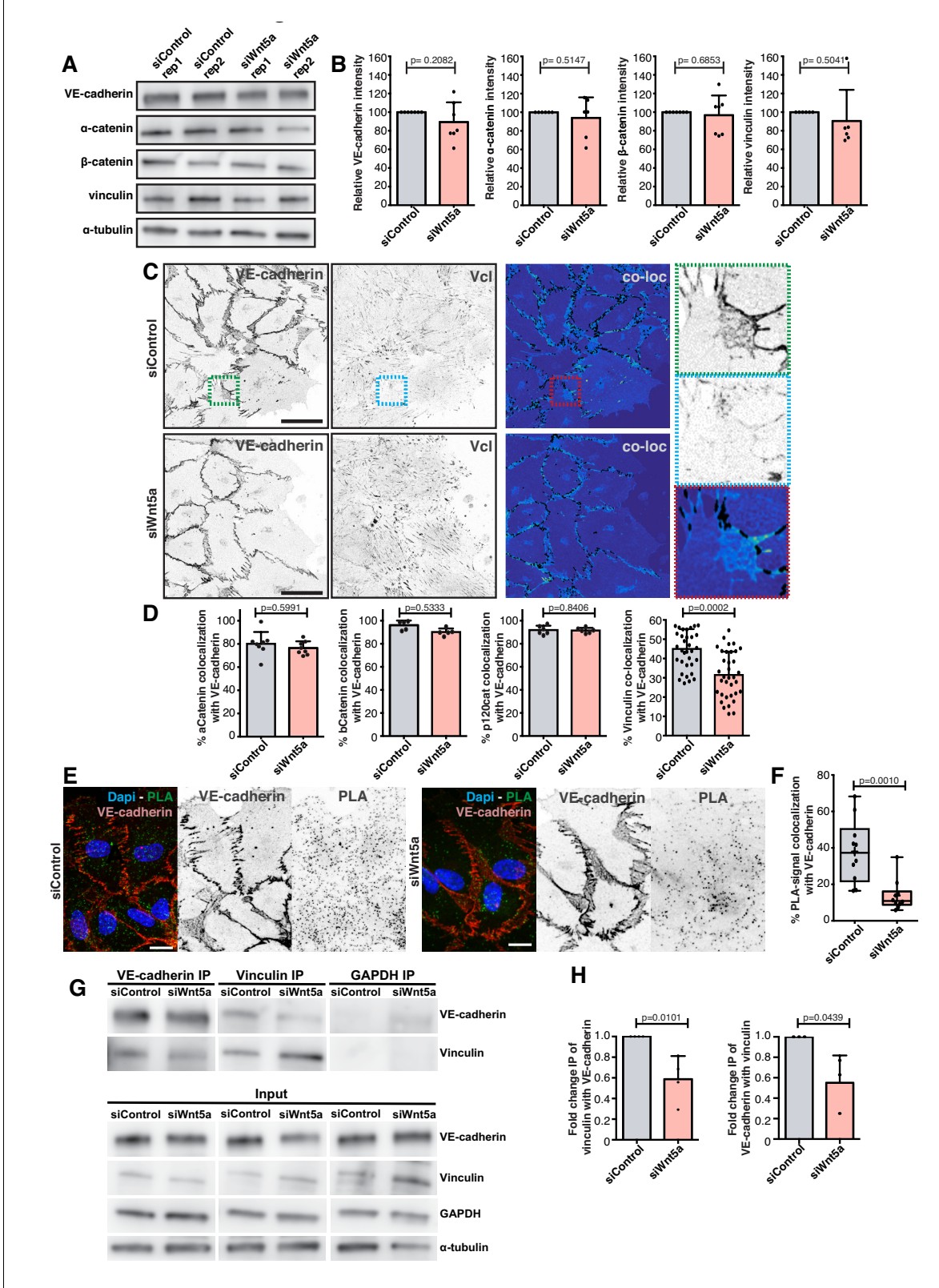

**Figure 4.** Wnt5a signaling promotes association of vinculin to the adherens junction complex. (**A**) Western blot for VE-cadherin, vinculin, α-catenin and β-catenin and α-tubulin in siControl and siWNT5a transfected cells. (**B**) Quantification of VE-cadherin, vinculin, α-catenin and β-catenin relative protein levels normalized to α-tubulin. Data are mean ± SD, p-values from unpaired t-test (n = 5 independent experiments). (**C**) Representative images of HUVECs close to the wound labeled for VE-cadherin and vinculin used for co-localization analysis in siControl (top left) and siWNT5a (bottom left)

*Figure 4 continued on next page*

*Figure 4 continued*

transfected cells and the corresponding segmentation image showing the co-localizing pixels between both stainings in black (top and bottom right). Green (top right), blue (middle right) and red (bottom right) squares show a higher magnification of a junction where VE-cadherin and vinculin co-localize. Scale bar, 40 µm. (D) Co-localization (%) between α-catenin/VE-cadherin (n = 8 images, from three independent experiments), β-catenin/VE-cadherin (n = 5 images, from two independent experiments), p120Catenin/VE-cadherin (n = 6 images, from two independent experiments), and vinculin/VE-cadherin (n = 39 images, from six independent experiments) in siControl and siWNT5a transfected cells. Data are mean ± SD, p-values from unpaired t-test. (E) Representative images of HUVECs close to the wound labeled with VE-cadherin used for proximity ligation assay (PLA) between vinculin and VE-cadherin in siControl and siWNT5a transfected cells. Nucleus labeled with Dapi. Scale bar, 20 µm. (F) Co-localization (%) between PLA signal and VE-cadherin in siControl (n = 12 images, six independent experiments) and siWNT5a (n = 12 images, six independent experiments) transfected cells. Data are mean ± SD, p-values from unpaired t test. (G) VE-cadherin (n = 3) and vinculin (n = 4) co-immunoprecipitation in siControl and siWNT5a transfected cells. GAPDH co-immunoprecipitation was used as a control. (H) Fold change quantification of vinculin-VE-cadherin (n = 3) and VE-cadherin-vinculin (n = 4) binding in siControl and siWNT5a transfected cells. Data are mean ± SD, p-values from unpaired t test.

DOI: https://doi.org/10.7554/eLife.45853.007

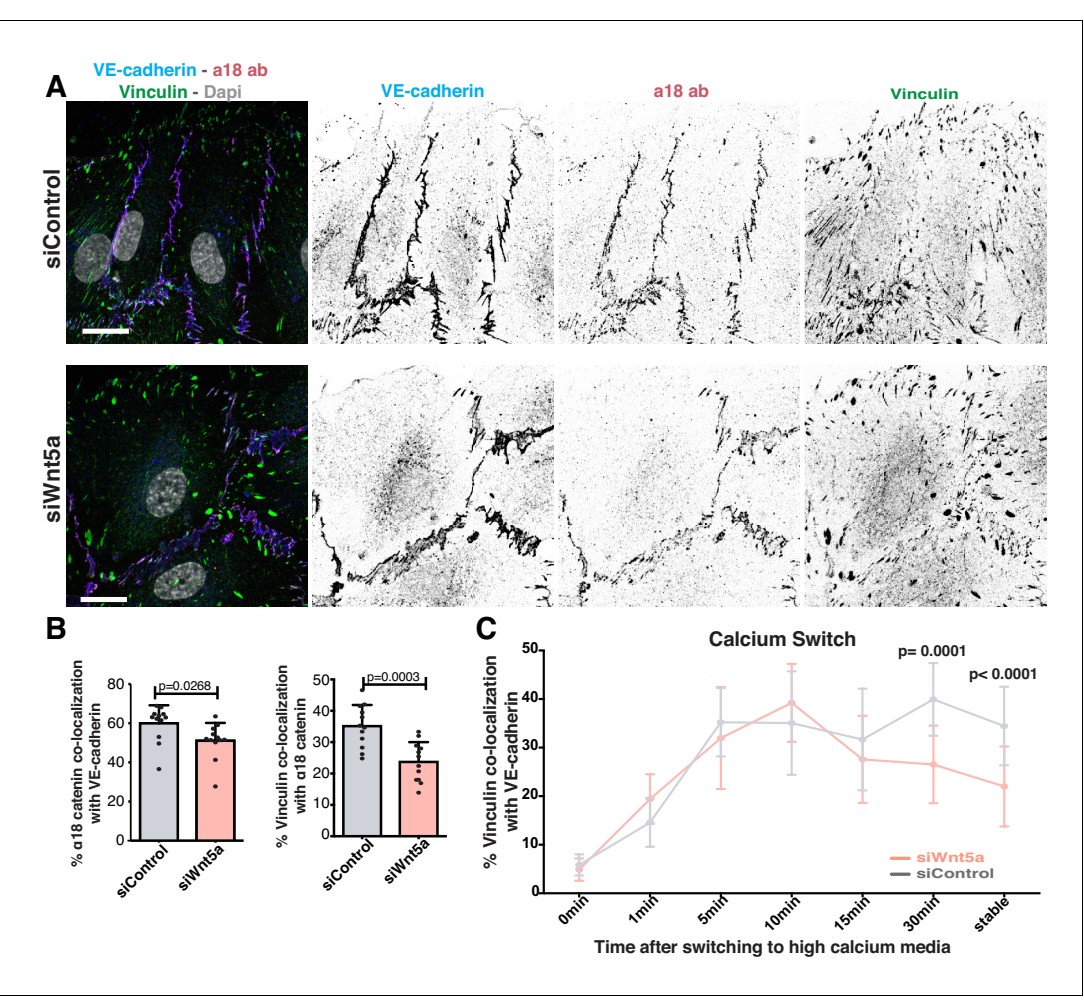

**Figure 5.** Wnt5a signaling stabilizes vinculin at adherens junctions. (A) Representative images of HUVECs close to the wound stained for nuclei (Dapi), VE-cadherin (Cdh5), α18-catenin and vinculin for co-localization studies in siControl and siWNT5a transfected cells. Scale bar, 20 µm. (B) Co-localization (%) between α18-catenin/VE-Cadherin and vinculin/α18 catenin in siControl (n = 12 images, three independent experiments) and siWNT5a (n = 12 images, three independent experiments) transfected cells. Data are mean ± SD, p-values from unpaired t-test. (C) Co-localization (%) between vinculin/VE-cadherin as function of calcium incubation time (min) after the calcium switch in HUVECs monolayers of siControl and siWNT5a transfected cells. Data are mean ± SD, p-values from unpaired t-test (n = 9–15 images per time point per condition, from 2 to 3 independent experiments).

DOI: https://doi.org/10.7554/eLife.45853.008

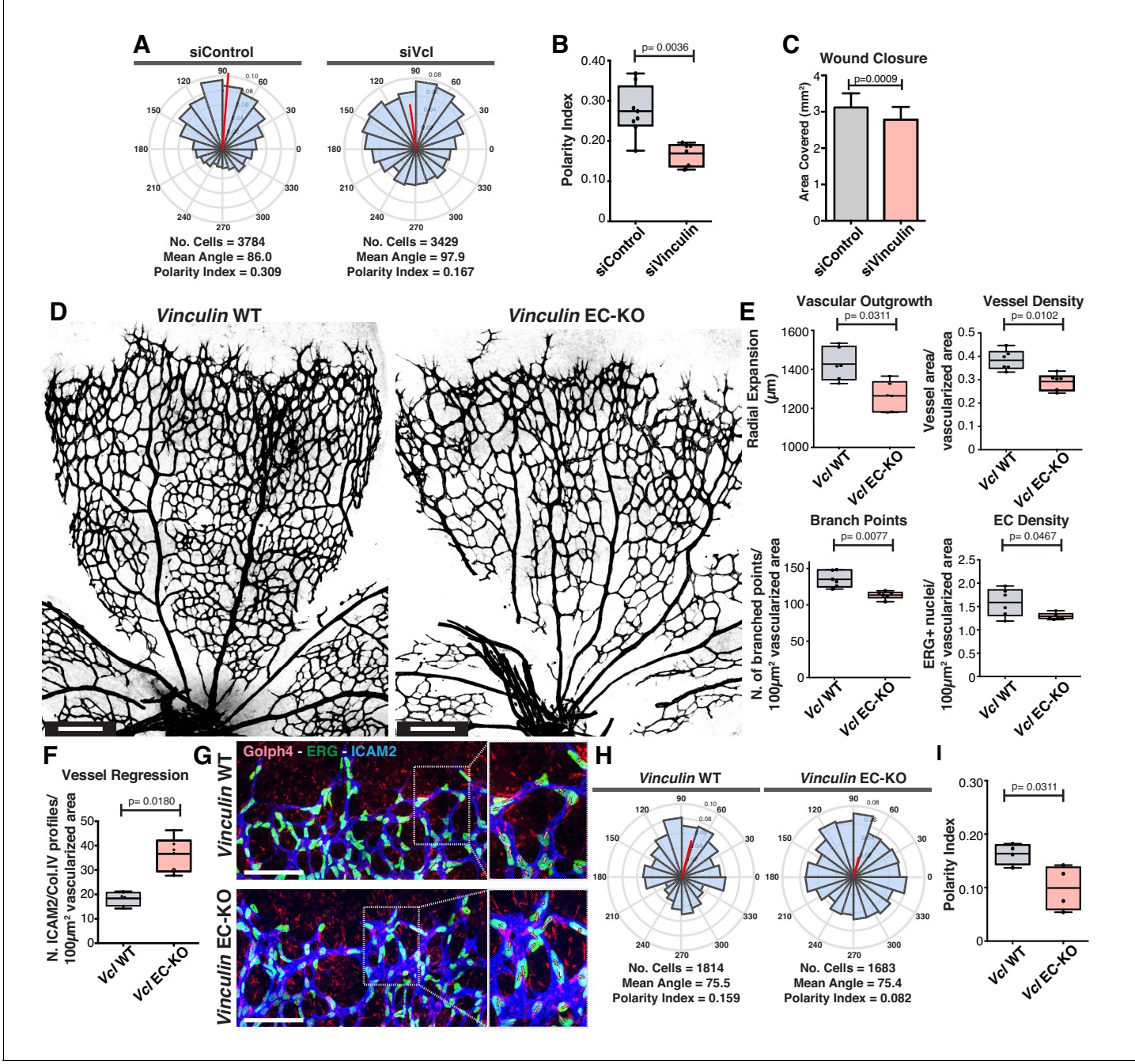

**Figure 6.** Vinculin is essential for sprouting angiogenesis and collective cell polarity. (A) Angular histograms showing the distribution of polarization angles from siControl (n = 10) and siVinculin (n = 11) transfected cells. (B) Polarity index box plots of leaders and followers from siControl (n = 8 images, from four independent experiments) and siVinculin (n = 6 images, from three independent experiments). p-values from unpaired t-test. (C) Quantification of wound closure over the course of 16 hr migration in siControl and siWNT5a transfected cells. N = 4 independent experiments. Data are mean ± SEM, p-values from unpaired t-test. (D) Representative images of overviews of mouse retinas from *Vinculin* WT and *Vinculin* EC-KO labeled for CD31. Scale bar, 250 μm. (E) Box plots of vascular outgrowth, vessel density, number of branch points and EC density in *Vinculin* WT (n = 6 retinas) and *Vinculin* EC-KO (n = 6 retinas) mouse retinas. p-values from unpaired t-test. (F) Box plot of vessel regression events in *Vinculin* WT (n = 4 retinas) and *Vinculin* EC-KO (n = 6 retinas) mouse retinas. p-values from unpaired t-test. (G) Representative images of sprouting fronts from *Vinculin* WT and *Vinculin* EC-KO; mouse retinas labeled for EC nuclei (Erg, green), lumen (Icam2, blue) and Golgi (Golph4, red). Each insert shows corresponding image segmentation of the vascular plexus showing axial polarity vectors. Scale bar, 200 μm. (H) Angular histograms showing the distribution of polarization angles of endothelial cells at the vascular sprouting front from *Vinculin* WT (n = 4 retinas) and *Vinculin* EC-KO (n = 4 retinas) mouse retinas. (I) Polarity index box plots of endothelial cells at the vascular sprouting front from *Vinculin* WT (n = 4 retinas) and *Vinculin* EC KO (n = 4 retinas) mouse retinas. p-values from unpaired t-test.

DOI: https://doi.org/10.7554/eLife.45853.009

(*Figure 6G–I*). Altogether, these results indicate that *Vinculin* is necessary for efficient collective cell polarity in endothelial cells in vitro and in vivo. Remarkably, the *Vinculin* EC-KO phenotype shows strong similarities with the one reported for non-canonical Wnt signaling EC-KO not only in terms of radial expansion, vessel density and regression profiles (*Franco et al., 2016*), but also in terms of polarity patterns (*Figure 1M,N*), suggesting that *Vinculin* might participate in a pathway regulated by non-canonical Wnt signaling.

## Constitutively active vinculin is sufficient to rescue collective behavior defects in Wnt5a-deficient endothelial cells

Our cumulative observations place junctional vinculin as the main mediator of Wnt5a signaling in collective cell behavior. This prompted us to test whether reinstating junctional vinculin activity would rescue Wnt5a deficiency. To this end, we overexpressed either full-length chicken vinculin (Vinc-FL) or chicken vinculin T12 (Vinc-T12) in siControl and siWNT5a cells. Vinc-T12 carries four amino acid mutations in its protein sequence which weaken the affinity of the auto-inhibitory head-to-tail interaction by 100-fold (*Cohen et al., 2005*). Thus, Vinc-T12 is considered to be a constitutively active vinculin. We confirmed that both constructs were able to efficiently rescue polarity defects of siVinculin cells (*Figure 7—figure supplement 1*). Overexpression of either form of vinculin did not affect significantly the strength of polarity of control cells (*Figure 7A,B*). Remarkably, Vinc-T12 but not Vinc-FL rescued impaired polarity of Wnt5a KD cells (*Figure 7A,B*, and *Figure 7—figure supplement 2*). Furthermore, overexpression of Vinc-T12 but not Vinc-FL led to a rescue in the organization of junctions in siWNT5a cells, promoting the formation of serrated high-tension junctions with the concomitant decrease in reticular junctions (*Figure 7C,D*). To confirm if vinculin's actin binding properties are required downstream of Wnt5a signaling pathway, we overexpressed a fusion protein containing the β-catenin binding domain of α-catenin and the actin-binding domain of vinculin (*Figure 8A*) (*Maddugoda et al., 2007*). α-catenin-vinculin (αCat-Vinc) fusion protein strongly localizes to adherens junctions (*Figure 8B*). αCat-Vinc overexpression did not significantly affect the overall PI of control cells, whilst it completely rescued collective cell polarity defects in siWNT5a cells (*Figure 8C,D* and *Figure 8—figure supplement 1*). Moreover, αCat-Vinc overexpression was sufficient to rescue cell migration straightness, the ratio of displacement to trajectory length, in siWNT5a cells (*Figure 8E*). Altogether, these observations are highly indicative that Wnt5a signaling leads to the activation of vinculin at adherens junctions to promote stable interactions between α-catenin and the actin cytoskeleton.

## Non-canonical Wnt signaling regulates junctional vinculin activity and collective cell polarity through the ROR2-Cdc42 signaling axis

To investigate how Wnt5a signaling leads to vinculin activity at the junctions, we screened for cell polarity defects upon downregulation of several known receptors for non-canonical Wnt ligands. Of all receptors tested, siROR2 was the only one phenocopying WNT5A depletion (*Figure 9A,B* and *Figure 9—figure supplement 1*). Moreover, siROR2 cells also showed a significant decrease in VE-cadherin-vinculin co-localization (*Figure 9C*). ROR2 is a tyrosine kinase receptor and it has been shown to activate JNK, Rac1 and Cdc42 pathways downstream of Wnt5a stimulation (*Green et al., 2014*; *Lee and Heur, 2014*; *Schambony and Wedlich, 2007*; *Stricker et al., 2017*). Inhibition of Rac1 or JNK did not affect collective cell polarity (*Figure 9D*). However, inhibition or siCdc42 impaired collective polarity of endothelial cells (*Figure 9D–F*). In accordance, siCdc42 impaired vinculin co-localization with VE-cadherin (*Figure 9G*). Analogous to siWNT5a, siCdc42 showed a significant decrease in the number of high-force serrated junctions (*Figure 9H,I*), and a significant reduction in the association between actin stress fibers and VE-cadherin (*Figure 9J*). PAK1-PBD-mediated pull-down of active GTP-bound Cdc42 confirmed that Wnt5a activates Cdc42 via ROR2 (*Figure 9K*). Moreover, using a FRET sensor of active Cdc42 (Cdc42-2G) (*Martin et al., 2016*), we observed activation of Cdc42 at cell-cell boundaries in siControl cells (*Figure 9L,M*, *Videos 1* and *2*). Interestingly, siWNT5a cells showed a significant decrease in the number of Cdc42-activation peaks at cell junctions between leader-follower or follower-follower cells when compared to siControl cells, whilst activation at the leading edge of leader cells was comparable between conditions (*Figure 9N* and *Videos 3* and *4*). To test whether Cdc42 regulates collective cell polarity during sprouting angiogenesis in vivo (*Laviña et al., 2018*), we inhibited Cdc42 activity in postnatal mouse

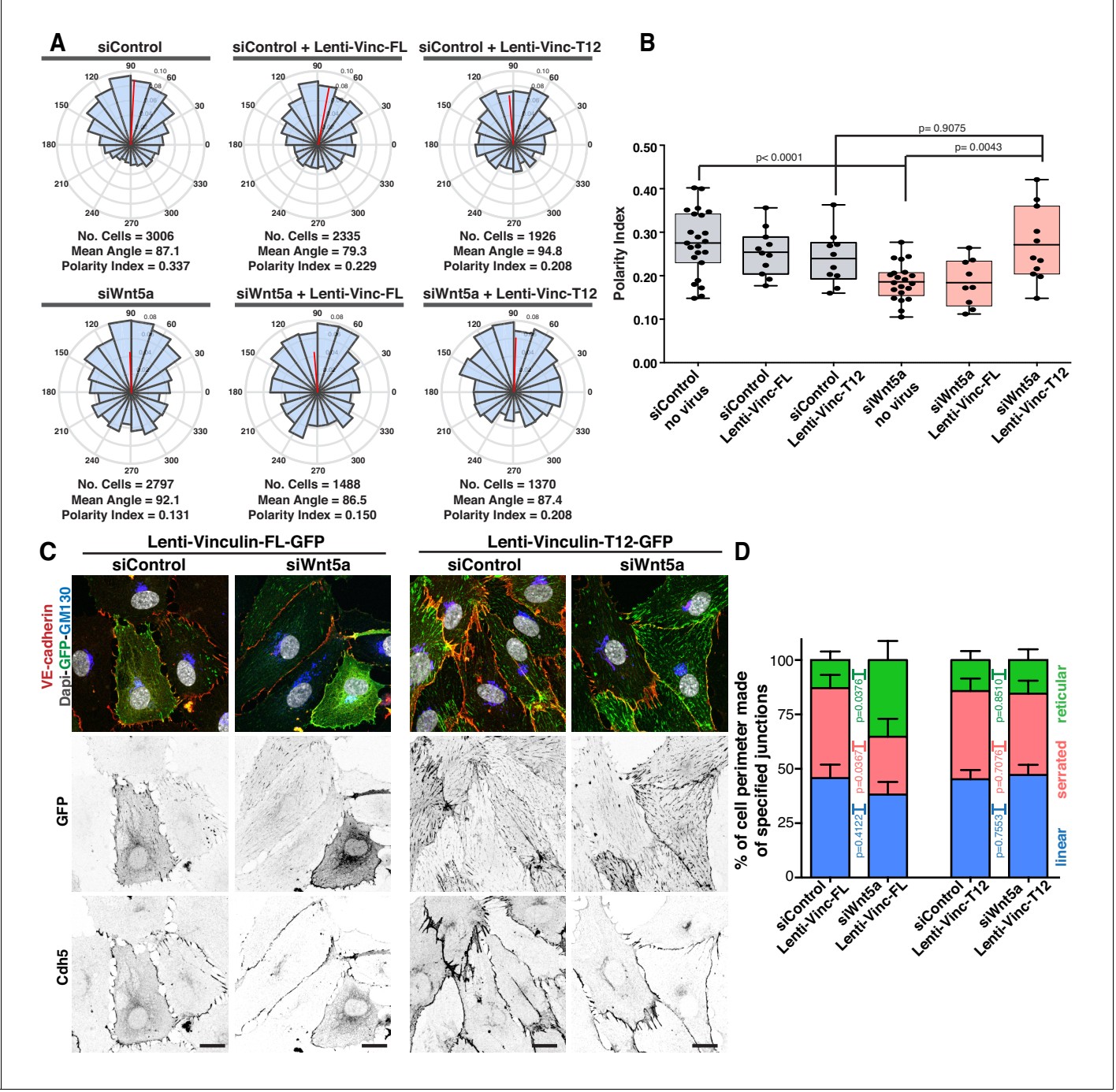

**Figure 7.** Active vinculin rescues Wnt5a deficiency. (A) Angular histograms showing the distribution of polarization angles from siControl and siWNT5a transfected cells either non-infected (n = 21–23 images, from six independent experiments) or expressing Vinculin-Full-Length-GFP (n = 9–11 images, from six independent experiments) or Vinculin-T12-GFP (n = 9–11 images, from six independent experiments). (B) Polarity index box plots of siControl and siWNT5a transfected cells either non-infected (n = 21–23 images, from six independent experiments) or Vinculin-Full-Length-GFP (n = 9–11 images, from six independent experiments) or Vinculin-T12-GFP (n = 9–11 images, from six independent experiments). p-values from unpaired t-test. (C) siControl and siWNT5a transfected HUVECs expressing Vinculin-Full-Length-GFP and Vinculin-T12-GFP. Nucleus labeled with Dapi, Golgi apparatus with GM130 and adherens junctions with VE-Cadherin. Scale bar, 20 μm. (D) Quantification of cell perimeter (%) composed of linear (blue), serrated (red) and reticular (green) in siControl and siWNT5a transfected cells expressing either Vinculin-Full-Length-GFP (n = 16 and 10 cells, respectively, from three independent experiments) or Vinculin-T12-GFP (n = 20 and 21 cells, respectively, from three independent experiments). Data are mean ± SD and p-values from unpaired t-test.

DOI: https://doi.org/10.7554/eLife.45853.010

*Figure 7 continued on next page*

*Figure 7 continued*

The following figure supplements are available for figure 7:

**Figure supplement 1.** Overexpression of exogenous vinculin isoforms rescues polarity defects of vinculin siRNA depleted cells.
DOI: https://doi.org/10.7554/eLife.45853.011

**Figure supplement 2.** Expression levels of exogenous vinculin isoforms in siControl and siWNT5a cells.
DOI: https://doi.org/10.7554/eLife.45853.012

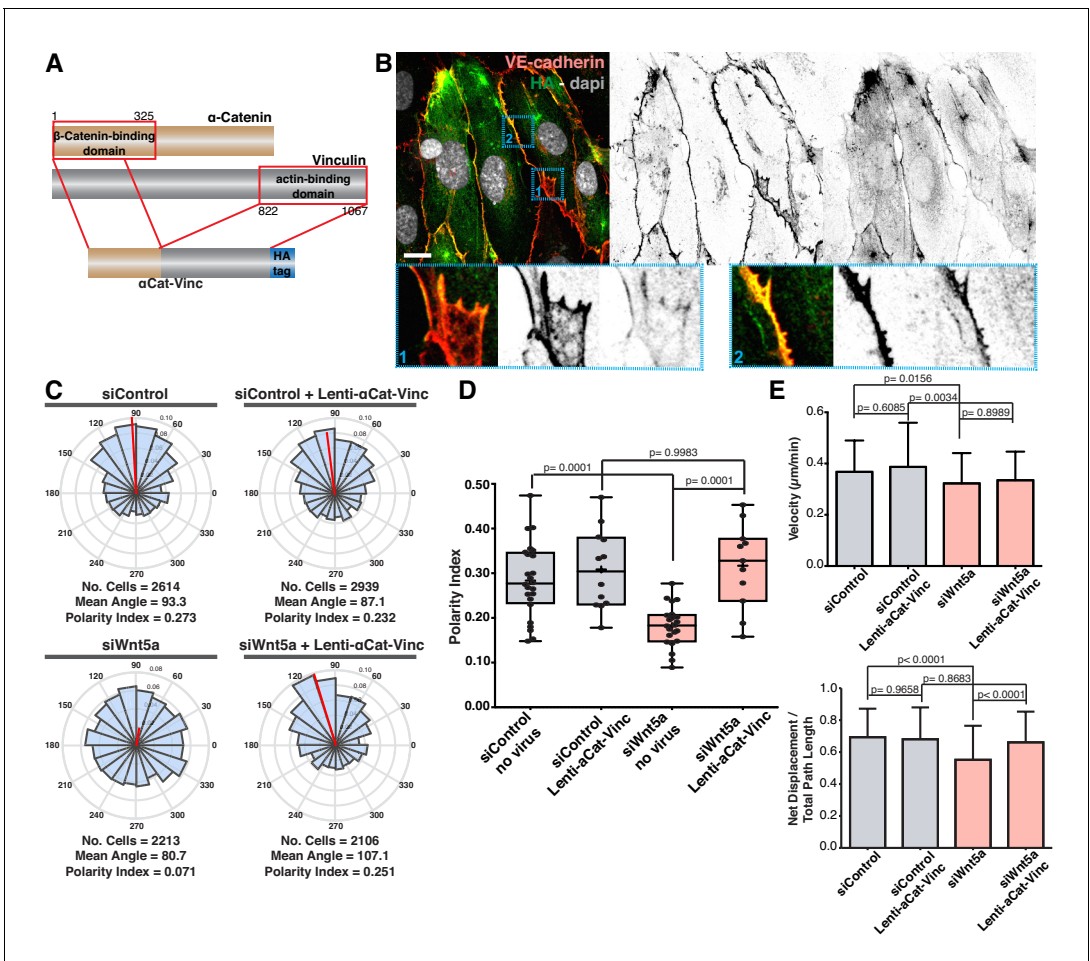

**Figure 8.** Forced vinculin binding to a-catenin rescues Wnt5a KD phenotype. (A) Diagram showing the molecular structure of the αCat-Vinc construct. αCat-Vinc-HA is a fusion protein containing the β-catenin-binding domain of α-catenin (brown) fused with the actin-binding domain of vinculin (gray) and the HA tag (blue). (B) Example of HUVECs expressing αCat-Vinc-HA. Nucleus labeled with Dapi, adherens junctions with VE-Cadherin. Scale bar, 20 μm. Blue square 1 (bottom left) shows a higher magnification of a reticular junction where HA does not co-localize with VE-Cadherin. Blue square 2 (bottom right) shows a higher magnification of a linear junction where HA and VE-cadherin co-localize. (C) Angular histograms showing the distribution of polarization angles from siControl and siWNT5a cells either non-infected (n = 22–24 images, from six independent experiments) or expressing αCat-Vinc-HA (n = 11–12 images, from six independent experiments). (D) Polarity index box plots of siControl and siWNT5a cells either non-infected (n = 22–24 images, from six independent experiments) or expressing αCat-Vinc-HA (n = 11–12 images, from six independent experiments). p-values from unpaired t-test. (E) Quantification of cell velocity and straightness over the course of 16 hr migration in siControl and siWNT5a transfected cells either non-infected (n = 150 cells, from three independent experiments) or expressing αCat-Vinc-HA (n = 150 cells, from three independent experiments). Data are mean ± SEM, p-values from unpaired t test compare siControl and siWNT5a groups.
DOI: https://doi.org/10.7554/eLife.45853.013

The following figure supplement is available for figure 8:

**Figure supplement 1.** Overexpression of a-catenin-vinculin fusion protein rescues polarity defects of Wnt5a siRNA depleted cells.
DOI: https://doi.org/10.7554/eLife.45853.014

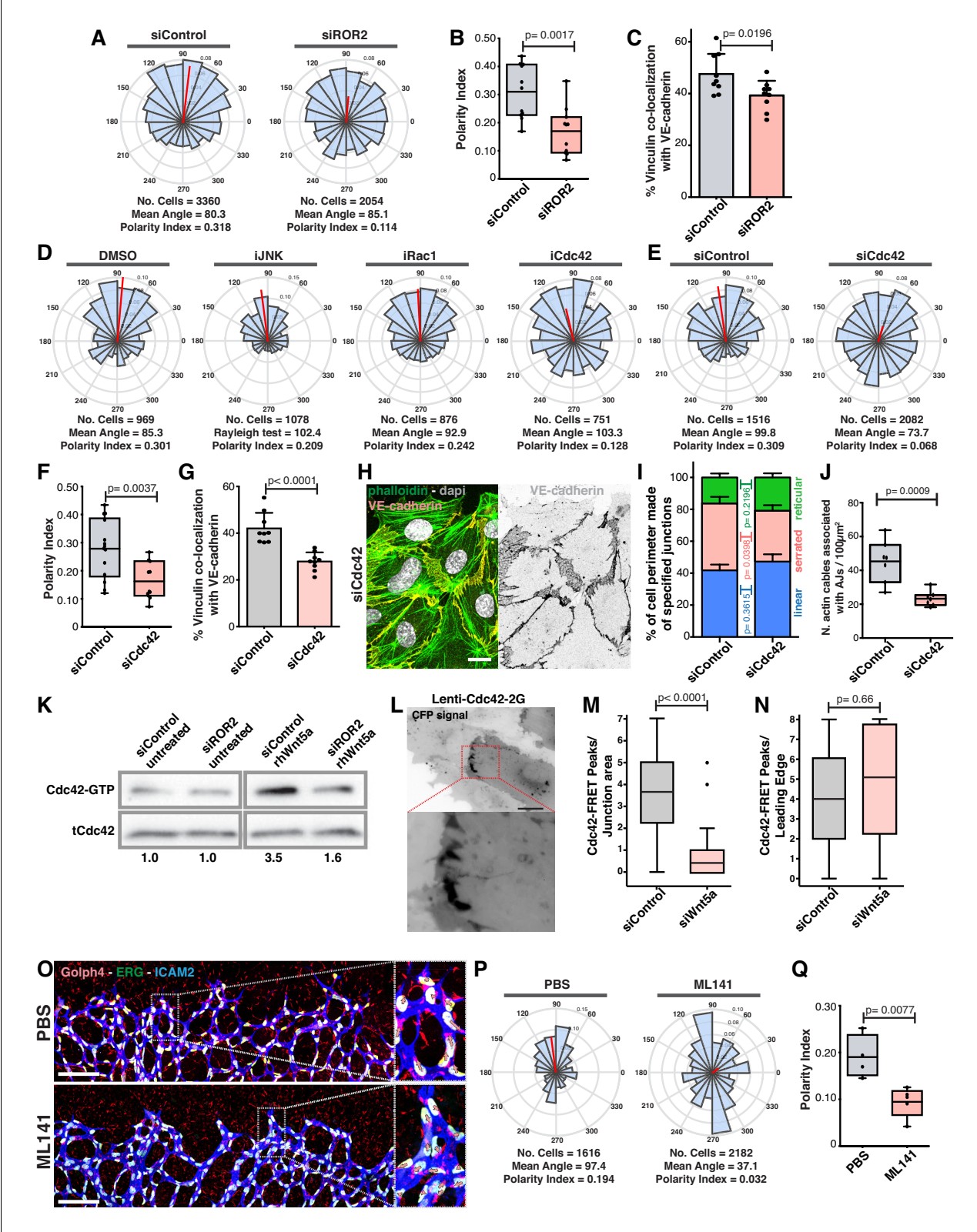

**Figure 9.** Wnt5a stabilizes vinculin at adherens junctions through a ROR2/Cdc42 pathway. (**A**) Angular histograms showing the distribution of polarization angles from siControl (n = 11 images, from six independent experiments) and siROR2 (n = 11 images, from six independent experiments) transfected cells. (**B**) Polarity index box plots from siControl (n = 11 images, from six independent experiments), and siROR2 (n = 11 images, from six independent experiments) cells. p-values from unpaired t-test. (**C**) Co-localization (%) between Vinculin/VE-Cadherin in siControl and siROR2

*Figure 9 continued on next page*

*Figure 9 continued*

transfected cells (n = 9 images, from three independent experiments). Data are mean ± SD, p-values from unpaired t-test. (D) Angular histograms showing the distribution of polarization angles of followers from wild type cells treated with either DMSO, iJNK (SP600125), iRac (NSC27632) or iCdc42 (ML141). n = 4 images, from two independent experiments. (E) Angular histograms showing the distribution of polarization angles from siControl (n = 14 images, from five independent experiments) and siCdc42 (n = 11 images, from five independent experiments) transfected cells. (F) Polarity index box plots from siControl (n = 14 images, from five independent experiments) and siCdc42 (n = 11 images, from five independent experiments) transfected cells. p-values from unpaired t-test. (G) Co-localization (%) between vinculin-VE-cadherin (n = 8 images, from three independent experiments) in siControl and siCdc42 transfected cells. Data are mean ± SD, p-values from unpaired t-test. (H) Detail of adherens junctions showing the association of actin stress fibers (phalloidin) to the adherens junctions (VE-Cadherin) of adjacent HUVECs in siCdc42 transfected cells. Nucleus labeled with Dapi. Scale bar, 20 µm. (I) Quantification of cell perimeter (%) composed of linear (blue), serrated (red) and reticular (green) in siControl and siCdc42 transfected cells (n = 78 and 75 cells, respectively, from two independent experiments). Data are mean ± SEM and p-values from unpaired t-test. (J) Quantification of the number of actin stress fibers connected to VE-cadherin positive cell-cell junctions in siControl or siCdc42 treated cells. N = 7 images, from three independent experiments. Data are mean ± SD, and p-values from unpaired t-test. (K) Pulldown of active GTP-bound Cdc42 in siControl and siROR2 transfected cells unstimulated or stimulated with recombinant human Wnt5a protein (rhWnt5a) (n = 1). (L) HUVEC expressing Cdc42-2G at adherens junction. Scale bar = 20 µm. (M) Box plots showing the number of Cdc42 FRET peaks per junction in siControl (n = 11 cell-cell interfaces, from two independent experiments) and siWNT5a (n = 9 cell-cell interfaces, from two independent experiments) transfected cells. p-values from unpaired t-test. (N) Box plots showing the number of Cdc42 FRET peaks per leading edge in siControl (n = 5 leading edges, from two independent experiments) and siWNT5a (n = 6 leading edges, from two independent experiments) transfected cells. p-values from unpaired t-test. (O) Left: example of a mouse retina sprouting front treated with PBS and Ml141 labeled for EC nuclei (Erg, green), lumen (Icam2, blue) and Golgi (Golph4, red). Right: higher magnification of the sprouting front showing high cell polarity coordination in PBS treated retinas and poor cell polarity coordination in Ml141 treated retinas. Scale bar, 200 µm. (P) Angular histograms showing the distribution of polarization angles of endothelial cells at the vascular sprouting front from mouse retinas treated with PBS (n = 4 retinas) or Ml141 (n = 5 retinas). (Q) Polarity index box plots of endothelial cells from mouse retinas treated with PBS (n = 4 retinas) or Ml141 (n = 5 retinas). p-values from unpaired t-test.

DOI: https://doi.org/10.7554/eLife.45853.015

The following figure supplement is available for figure 9:

**Figure supplement 1.** Polarity Indexes of endothelial cells depleted on specific receptors related to non-canonical Wnt signaling.

DOI: https://doi.org/10.7554/eLife.45853.016

pups, as previously reported (*Fantin et al., 2015*), and quantified collective polarity of endothelial cells at the vascular sprouting front. Remarkably, inhibition of Cdc42 led to a specific and significant randomization of endothelial cell polarity at the angiogenic sprouting front in vivo (*Figure 9O–Q*). Thus, our results confirm that Cdc42 regulates collective cell polarity during sprouting angiogenesis in vivo.

Taken together, we propose that Wnt5a signaling, through ROR2-Cdc42 activity, stabilizes vinculin at adherens junctions to reinforce its connection to the actin cytoskeleton. In this context, non-canonical Wnt signaling reinforces mechanocoupling between endothelial cells, which is essential for collective cell polarity in sprouting angiogenesis.

## Discussion

Sprouting angiogenesis requires efficient coordination of cell specification, cell proliferation, cell migration, and cell rearrangements. Previous work has elucidated the basic cellular and molecular mechanisms leading to endothelial tip/stalk cell specification and proliferation (*Potente and Mäkinen, 2017*). Yet, the mechanisms controlling collective cell polarity, migration and cell rearrangements at the vascular sprouting front are still poorly understood. Here, we identify a novel signaling pathway that reinforces mechanocoupling between endothelial cells to

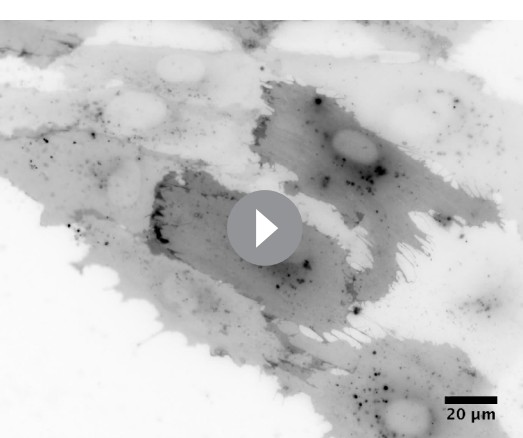

**Video 1.** Localization of Cdc42-FRET sensor in wounded monolayers. ECFP fluorescent signal from Cdc42-2G FRET sensor at the leading edge of siControl cells. Photobleaching effects were corrected using a FIJI plugin. Images were acquired for 5 min with 1 s time interval.

DOI: https://doi.org/10.7554/eLife.45853.017

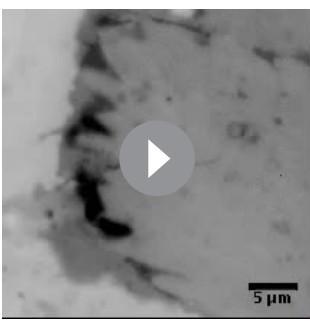

**Video 2.** Highlight of *Video 1*. Crop from *Video 1*, showing the ECFP fluorescent signal from the Cdc42-2G FRET sensor in an interface between leader and follower siControl cells. Photobleaching effects were corrected using a FIJI plugin. Images were acquired for 5 min with 1 s time interval.

DOI: https://doi.org/10.7554/eLife.45853.018

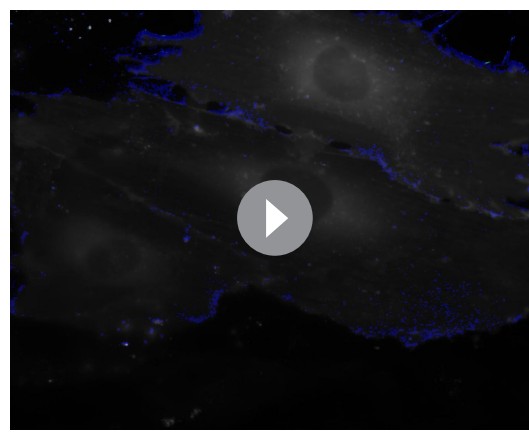

**Video 3.** Ratiometric FRET signal in Cdc42-2G sensor in siControl cells. Ratiometric FRET signal from Cdc42-2G (blue scale) from *Video 1*, superimposed to the acceptor signal (gray scale) in siControl cells. Photobleaching effects were not corrected. Images were acquired for 5 min with 1 s time interval.

DOI: https://doi.org/10.7554/eLife.45853.019

coordinate collective cell polarity and migration during sprouting angiogenesis. We uncover that Wnt5a, through ROR2, activates Cdc42 at adherens junctions, which is necessary for stable binding of vinculin to α-catenin, and efficient mechanocoupling between endothelial cells (*Figure 10*). Low non-canonical Wnt signaling weakens adherens junctions, impairs force propagation, and disrupts collective behavior of endothelial cells, which in turn affects angiogenic sprouting efficiency.

We identify that Cdc42 plays an important role downstream of Wnt5a-ROR2 signaling in the regulation of vinculin's stabilization and activation at adherens junctions. Cdc42 is a well-known regulator of cell polarity, playing important roles in yeast budding, epithelial polarity, migratory polarity and fate specification during cell division (*Heasman and Ridley, 2008*). In this context, Cdc42 frequently interacts with the PAR complex (PAR6–PAR3–aPKC) to mediate both front-rear polarity and apical-basal polarity (*Etienne-Manneville and Hall, 2001*; *Etienne-Manneville and Hall, 2003*; *Wu et al., 2007*). In endothelial cells, Cdc42 was previously implicated in filopodia formation (*Barry et al., 2015*; *Fantin et al., 2015*; *Wakayama et al., 2015*), adherence, junction stability (*Broman et al., 2006*), cell migration (*Vitorino and Meyer, 2008*; *Wakayama et al., 2015*), and more recently on collective polarity (*Laviña et al., 2018*). Yet, Cdc42 seems to be dispensable for apical-basal but essential for front-rear polarization (*Laviña et al., 2018*). Interestingly, non-canonical Wnt pathway was shown to cooperate with Cdc42/PAR complex to regulate front-rear polarity in migrating fibroblasts at the leading edge (*Schlessinger et al., 2007*), evoking two parallel mechanisms regulating polarity of leader cells. This fits with our own results, as leader cells were mildly affected by Wnt5a KD. In endothelial cells, we show that Wnt5a regulates Cdc42 activity at cell-cell boundaries but not at the leading edge of leader cells. This signaling spatial regulation could explain why leader cells are less affected by deficient Wnt5a signaling. Indeed, Cdc42 inhibition

**Video 4.** Ratiometric FRET signal in Cdc42-2G sensor in siWNT5a cells. Ratiometric FRET signal from Cdc42-2G (blue scale) superimposed to the acceptor signal (gray scale) in siWNT5a cells. Photobleaching effects were not corrected. Images were acquired for 5 min with 1 s time interval.

DOI: https://doi.org/10.7554/eLife.45853.020

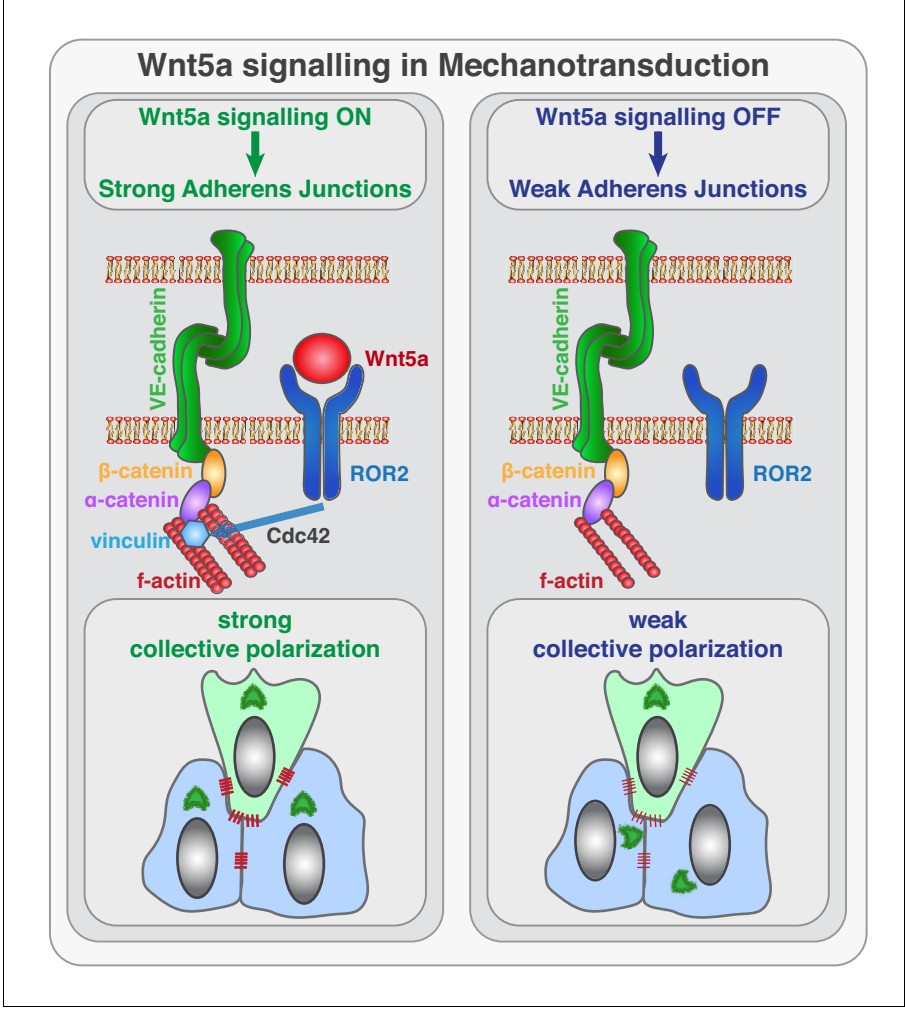

**Figure 10.** Schematic of the function of Wnt5a signaling in mechanocoupling at adherens junctions. Working model for the role of non-canonical Wnt ligand WNT5a in mechanotransduction. Wnt5a, through ROR2, activates Cdc42 at adherens junctions, which is necessary for stable binding of vinculin to α-catenin, and efficient mechanocoupling between endothelial cells. Low non-canonical Wnt signaling weakens adherens junctions, impairs force propagation, and disrupts collective cell migration of endothelial cells.
DOI: https://doi.org/10.7554/eLife.45853.021

or LOF in vitro or in vivo gives to a stronger polarity phenotype than non-canonical Wnt signaling LOF experiments (*Laviña et al., 2018*). This suggests that Cdc42 is regulated by multiple inputs to control cell polarity, and that Wnt5a signaling fine-tunes Cdc42 activity at cell-cell junctions.

Our data further shows that the role of non-canonical Wnt signaling on mechanocoupling relies on vinculin stabilization at adherens junctions. The biological function of vinculin at adherens junctions has been a theme of controversy. Despite being present at high-tension junctions in several model organisms, vinculin is dispensable for zebrafish and fruitfly normal development (*Alatortsev et al., 1997*; *Han et al., 2017*). However, its absence during mouse embryonic development results in lethal cardiovascular and neuronal defects (*Xu et al., 1998*). To explain these differences, it has been proposed that mechanical and molecular properties of proteins from the adherens junctions might have diverged during evolution (*Han et al., 2017*). For instance, zebrafish α-catenin is monomeric and can form a complex with β-catenin and F-actin simultaneously, whilst the murine orthologue forms dimers and cannot bind simultaneously to F-actin and β-catenin in solution (*Buckley et al., 2014*; *Miller et al., 2013*). Thus, vinculin is required to promote efficient coupling between α-catenin and F-actin in mouse. However, the factors that would regulate these

interactions are so far elusive. Our results are compatible with the idea that a Wnt5a-ROR2-Cdc42 signaling axis could have evolved in mammals to enhance cadherin mechanoproperties through vinculin. Moreover, the ability to rescue the collective cell polarity defects on Wnt5a-deficient cells by re-expression of Vinc-T12 or αCat-Vinc fusion protein further suggests that Wnt5a modulates mechanocoupling efficiency by regulating vinculin's actin-binding properties. How Wnt5a affects the dynamics or affinity of vinculin to actin filaments shall be investigated in future work.

In addition, our results strongly suggest that Wnt5a acts as a permissive rather than an instructive cue regarding cell polarity. The ability to rescue the collective polarity phenotype of siWNT5a cells by re-expression of either Vinc-T12 or αCat-Vinc fusion protein implies that the polarity cue organizing collective cell polarity does not depend on Wnt5a. In this context, Wnt5a seems to be mainly necessary to potentiate mechanocoupling between cells via vinculin activation, a condition sufficient to propagate the external polarity cue in the system. This is also concordant with our previous observations that overexpression of Wnt5a in endothelial cells in vivo led to normal vascular sprouting and remodeling phenotypes (*Franco et al., 2016*). Interestingly, a similar debate regarding a permissive or instructive role involves non-canonical Wnt signaling in planar cell polarity (PCP) establishment (*Humphries and Mlodzik, 2018*), where conflicting evidences exists in favor of each role. Our results suggest that non-canonical Wnt signaling plays a role in force transmission within cell populations. As mechanical cues were shown to play a contributing role in PCP establishment (*Humphries and Mlodzik, 2018*), a mechanobiology perspective into the function of non-canonical Wnt signaling in PCP could in part conciliate the possibility that non-canonical Wnt signaling can be seen as instructive or permissive, depending on the experimental setting.

Non-canonical Wnt signaling was previously implicated in the regulation of vessel regression (*Franco et al., 2016*; *Korn et al., 2014*). Intriguingly, *Vinculin* EC-KO shows a very similar phenotype, with an increase in vessel regression, and a decrease in vessel density and radial expansion. It was suggested that non-canonical Wnt signaling regulates vessel regression by controlling a mechanosensitive threshold, based on wall shear stress, that induces endothelial cell polarization and migration (*Franco et al., 2016*). The mechanisms controlling this threshold are still unclear. Given the well-known mechanoresponsive properties of vinculin, it is tempting to speculate that vinculin could also play a relevant role in establishing this threshold. Further work is necessary to clarify this question. Nevertheless, it is relevant to note that Wnt5a and vinculin regulates a different mechanosensitive pathway in flow-independent conditions. This also raises the question of how shear stress and junctional mechanotransduction pathways are regulated and coordinated by non-canonical Wnt signaling in space and time within the vascular network.

Taken together, our results show that Wnt5a signaling fine-tunes junctional mechanocoupling between endothelial cells to promote collective cell behavior during vascular morphogenesis.

# Materials and methods

## Key resources table

| Reagent type (species) or resource | Designation | Source or reference | Identifiers | Additional information |
|---|---|---|---|---|
| Strain, strain background (*Escherichia coli*) | Stbl3 | Life Technologies | Cat#: C7373-03 | Chemically Competent |
| Genetic reagent (*Mus musculus*) | *Vcl* fl/fl::*Pdgfb*-iCreERT2 | This paper | | Generated from *Vcl* floxed crossed with *Pdgfb*-iCreERT2 |
| Genetic reagent (*Mus musculus*) | *Vcl* fl/fl | (*Zemljic-Harpf et al., 2007*) | | Generated from *Vcl* floxed crossed with *Pdgfb*-iCreERT2 |
| Genetic reagent (*Mus musculus*) | *Pdgfb*-iCreERT2 | (*Claxton et al., 2008*) | | |
| Genetic reagent (*Mus musculus*) | *Wnt5a* fl/fl | (*Miyoshi et al., 2012*) | | |

*Continued on next page*

*Continued*

| Reagent type (species) or resource | Designation | Source or reference | Identifiers | Additional information |
|---|---|---|---|---|
| Genetic reagent (*Mus musculus*) | *Wnt11* null | (**Majumdar et al., 2003**) | | |
| Genetic reagent (*Mus musculus*) | *Wnt5a* fl/fl::Wnt11 null::*Pdgfb*-iCreERT2 | **Franco et al., 2016** | | |
| Cell line (*Homo sapiens*) | HEK293T | ATCC | ATCC:CRL3216; RRID:CVCL_0063 | |
| Cell line (*Homo sapiens*) | Human umbilical vein endothelial cells (HUVECs) | Lonza | Cat#: C2519A | Primary cell line |
| Antibody | AffiniPureF(ab') two fragments Donkey anti-rabbit IgG | Jackson ImmunoResearch | Cat#: 711-006-152; RRID:AB_2340586 | IF(1:400) |
| Antibody | Mouse anti-CD102 | BD Biosciences | Cat#: 553326; RRID:AB_394784 | IF(1:200) |
| Antibody | Rabbit anti-CDC42 | Cell Signaling | Cat#: 2466; RRID:AB_2078082 | WB(1:1000) |
| Antibody | Rabbit anti-Erg | Abcam | Cat#: ab92513; RRID:AB_2630401 | IF(1:200) |
| Antibody | Chicken anti-GFP | Aves Labs | Cat#: GFP-1010; RRID:AB_2307313 | WB(1:2000) |
| Antibody | Rabbit anti-GOLPH4 | Abcam | Cat#: ab28049; RRID:AB_732692 | IF(1:400) |
| Antibody | Mouse anti-HA tag | BioLegend | Cat#: 901513; RRID:AB_2565335 | IF(1:100), WB(1:500) |
| Antibody | Rabbit anti-p120-Catenin | Merck Millipore | Cat#: 05–1567; RRID:AB_11213674 | IF(1:100) |
| Antibody | Goat anti-VE-Cadherin | Santa Cruz Biotechnologies | Cat#: sc-6458; RRID:AB_2077955 | IF(1:100), WB(1:1000) |
| Antibody | Mouse anti-VE-Cadherin | Santa Cruz Biotechnologies | Cat#: sc-9989; RRID:AB_2077957 | IF(1:100) |
| Antibody | Goat anti-VE-Cadherin | R and D Systems | Cat#: AF938; RRID:AB_355726 | IF(1:100), WB(1:400) |
| Antibody | Mouse anti-Vinculin | Sigma-Aldrich | Cat#: V9264; RRID:AB_10603627 | IF(1:400), WB(1:400) |
| Antibody | Rabbit anti-Vinculin | Sigma-Aldrich | Cat#: V4139; RRID:AB_262053 | IF(1:100), WB(1:400) |
| Antibody | Rabbit anti-ZO1 | Invitrogen | Cat#: 402300; RRID:AB_2533457 | IF(1:100) |
| Antibody | Rat anti-α18 | Prof. Dr. Masatoshi Takeichi (RIKEN, Kobe) shared resource | | IF(1:20000) |
| Antibody | Rabbit anti-α-Catenin | Sigma-Aldrich | Cat#: C2081; RRID:AB_476830 | IF(1:200), WB(1:1000) |
| Antibody | Mouse anti-α-Tubulin | Sigma-Aldrich | Cat#: T6199; RRID:AB_477583 | IF(1:200), WB(1:2000) |
| Antibody | Rabbit anti-β-Catenin | Sigma-Aldrich | Cat#: C2206; RRID:AB_476831 | IF(1:100), WB(1:1000) |
| Antibody | Mouse anti-γ-Tubulin | Sigma-Aldrich | Cat#: T6557; RRID:AB_477584 | WB(1:2000) |
| Antibody | Donkey anti-Chicken HRP | Jackson ImmunoResearch | Cat#: 703-035-155; RRID:AB_10015283 | WB(1:5000) |
| Antibody | Donkey anti-Goat Alexa 647 | Thermo Fisher Scientific | Cat#: A21447; RRID:AB_2535864 | IF(1:400) |

*Continued on next page*

*Continued*

| Reagent type (species) or resource | Designation | Source or reference | Identifiers | Additional information |
|---|---|---|---|---|
| Antibody | Donkey anti-Goat HRP | Bethyl | Cat#: A50-201P; RRID:AB_66756 | WB(1:5000) |
| Antibody | Donkey anti-Mouse Alexa 488 | Thermo Fisher Scientific | Cat#: A21202; RRID:AB_141607 | IF(1:400) |
| Antibody | Donkey anti-Rabbit Alexa 568 | Thermo Fisher Scientific | Cat#: A10042; RRID:AB_2534017 | IF(1:400) |
| Antibody | Donkey anti-Rabbit Alexa 488 | Thermo Fisher Scientific | Cat#: A21206; RRID:AB_2535792 | IF(1:400) |
| Antibody | Donkey anti-Rabbit Alexa 647 | Thermo Fisher Scientific | Cat#: A21447; RRID:AB_2535864 | IF(1:400) |
| Antibody | Goat anti-Rabbit HRP | Life Technologies | Cat#: G-21234 | WB(1:5000) |
| Antibody | Goat anti-Rat Alexa 555 | Thermo Fisher Scientific | Cat#: A21434; RRID:AB_2535855 | IF(1:400) |
| Antibody | Phalloidin 488 | Thermo Fisher Scientific | Cat#: A12379 | IF(1:400) |
| Antibody | Phalloidin 568 | Thermo Fisher Scientific | Cat#: A12380 | IF(1:200) |
| Antibody | Sheep anti-Mouse HRP | GE Healthcare | Cat#: NA931V | WB(1:5000) |
| Recombinant DNA reagent | pLenti-Cdc42-2G | Prof. Dr. Olivier Pertz (Institute of Cell Biology) shared resource | Addgene plasmid #68813; RRID:Addgene_68813 | Lentiviral vector expressing a FRET sensor of active Cdc42 |
| Recombinant DNA reagent | Lifeact-mCherry | Prof. Dr. Edgar Gomes (Instituto de Medicina Molecular) shared resource | | |
| Recombinant DNA reagent | VE-Cad-TL | Prof. Martin Schwartz (Yale University) shared resource | Addgene plasmid #45849 pLPCX-VEcadTL; RRID:Addgene_45849 | Lentiviral vector expressing a FRET VE-cadherin tailless tension sensor |
| Recombinant DNA reagent | VE-Cad-TS | Prof. Martin Schwartz (Yale University) shared resource | Addgene plasmid #45848 pLPCX-VEcadTS; RRID:Addgene_45848 | Lentiviral vector expressing a FRET VE-cadherin tension sensor |
| Recombinant DNA reagent | Vinculin-Full Length-GFP | This paper | Addgene plasmid #46265 pEGFPC1/ GgVcl 1–1066 | Lentiviral vector expressing vinculin full-length tagged with GFP |
| Recombinant DNA reagent | Vinculin-T12 mutant-GFP | This paper | Addgene plasmid #46266 pEGFPC1/ GgVcl 1–1066 T12 mutant; RRID:Addgene_46266 | Lentiviral vector expressing vinculin T12 mutant tagged with GFP |
| Recombinant DNA reagent | αCat-Vinc-HA | This paper | Cloned in pUC57, General Biosytems | Lentiviral vector expressing a fusion protein containing the β-catenin binding domain of α-catenin and the actin-binding domain of vinculin tagged with HA |
| Recombinant DNA reagent | Wnt5a-V5 | This paper | | Lentiviral vector expressing Wnt5a tagged with V5 |
| Recombinant DNA reagent | pLX303 | Addgene | Cat#: 25897; RRID:Addgene_25897 | Lentiviral backbone |
| Sequence-based reagent | RT-qPCR primers | This paper | | See *Table 2* |

*Continued on next page*

*Continued*

| Reagent type (species) or resource | Designation | Source or reference | Identifiers | Additional information |
|---|---|---|---|---|
| Sequence-based reagent | ON-TARGET human siRNAs | Dharmacon | | See *Table 1* |
| Peptide, recombinant protein | Recombinant human Wnt5a protein | R and D Systems | Cat#: 645-WN | |
| Commercial assay or kit | BCA protein assay kit | VWR | Cat#: 786–0000 | |
| Commercial assay or kit | Cdc42 Pull-down Activation Assay Biochem Kit | Cytoskeleton | Cat#: BK034 | |
| Commercial assay or kit | Duolink In Situ Red Mouse/Rabbit Starter Kit | Sigma-Aldrich | Cat#: DUO92101 | |
| Commercial assay or kit | ECL Western Blotting Detection Reagent | GE Healthcare | Cat#: RPN2232 | |
| Commercial assay or kit | GeneJet RNA Purification Kit | Thermo Fisher Scientific | Cat#: K0731 | |
| Commercial assay or kit | RNeasy Mini Kit | Qiagen | Cat#: 74104 | |
| Commercial assay or kit | Superscript IV First-Strand Synthesis System | Invitrogen | Cat#: 18091050 | |
| Chemical compound, drug | Dabco (1,4-Diazabicyclo[2.2.2]octane) | Sigma-Aldrich | Cat#: D27802 | |
| Chemical compound, drug | DharmaFECT one reagent | Dharmacon | Cat#: T-2001–02 | |
| Chemical compound, drug | DNase I | NZYTech | Cat#: MB19901 | |
| Chemical compound, drug | DSP (dithiobis (succinimidyl propionate)) | Alfagene | Cat#: 22585 | |
| Chemical compound, drug | Fibronectin | Sigma-Aldrich | Cat#: F1141 | |
| Chemical compound, drug | Full Range Rainbow Recombinant protein Molecular weight marker | GE Healthcare | Cat#: RPN800E | |
| Chemical compound, drug | Gelatin | Sigma-Aldrich | Cat#: G1393 | |
| Chemical compound, drug | ML-141 | Sigma-Aldrich | Cat#: SML0407 | 10 µM |
| Chemical compound, drug | Mowiol | Sigma-Aldrich | Cat#: 81381 | |
| Chemical compound, drug | NSC 23766 | Tocris | Cat#: 2161 | 100 µM |
| Chemical compound, drug | Phosphatase and proteinase inhibitors cocktail | Thermo Fisher Scientific | Cat#: 1861281 | |
| Chemical compound, drug | Pierce G-protein agarose beads | Thermo Fisher Scientific | Cat#: 22851 | |
| Chemical compound, drug | Ponceau Red | NZYTech | Cat#: MB19201 | |

*Continued on next page*

*Continued*

| Reagent type (species) or resource | Designation | Source or reference | Identifiers | Additional information |
|---|---|---|---|---|
| Chemical compound, drug | Power SYBR Green PCR Master Mix | Thermo Fisher Scientific | Cat#: 4368706 | |
| Chemical compound, drug | SP600125 | Tocris | Cat#: 1496 | 10 µM |
| Chemical compound, drug | Tamoxifen | Sigma-Aldrich | Cat#: H7904 | |
| Chemical compound, drug | 4x Laemmli Sample Buffer | Bio-Rad | Cat#:161–0747 | |
| Software, algorithm | Adobe photoshop | Adobe Photoshop (https://www.adobe.com/products/photoshop.html) | RRID:SCR_014199 | Version CS4 |
| Software, algorithm | Biosensor Processing | (*Hodgson et al., 2010*) | | Version 2.1 |
| Software, algorithm | Cell image velocimetry (CIV) | (*Milde et al., 2012*) | | |
| Software, algorithm | Chemotaxis and Migration Tool | Chemotaxis and Migration Tool (https://ibidi.com/chemotaxis-analysis/171-chemotaxis-and-migration-tool.html) | | Version 2.0 |
| Software, algorithm | GraphPad Prism | GraphPad Prism (https://graphpad.com) | RRID:SCR_015807 | Version 7 |
| Software, algorithm | ImageJ | ImageJ (http://imagej.nih.gov/ij/) | RRID:SCR_003070 | |
| Software, algorithm | Image Lab | Image Lab (http://www.bio-rad.com/en-us/sku/1709690-image-lab-software) | RRID:SCR_014210 | Version 6.0.1 |
| Software, algorithm | Matlab script used for immunostaining co-localization analysis | This paper | | |
| Software, algorithm | Matlab script used for automated polarity analysis | This paper | | Modified version of polarity analysis script from Dr. Anne-Clémence Vion and Dr. Holger Gerhardt (Max-Delbruck Center) |
| Software, algorithm | Matlab script used for FRET analysis | This paper | | |
| Software, algorithm | Matlab script used for statistical analysis | Matlab (http://www.mathworks.com/products/matlab/) | RRID:SCR_001622 | |
| Software, algorithm | MetaMorph | MetaMorph (http://www.moleculardevices.com/Products/Software/Meta-Imaging-Series/MetaMorph.html) | RRID:SCR_002368 | |
| Software, algorithm | Velocity spatial correlation | (*Petitjean et al., 2010*) | | |
| Software, algorithm | Zen | Zen (http://www.zeiss.com/microscopy/en_us/products/microscope-software/zen.html#introduction) | RRID:SCR_013672 | |
| Other | DAPI stain | Sigma-Aldrich | Cat#: D9542 | |

**Table 1.** List of siRNAs.

| Name | Brand | Catalog number | Sequence |
|------|-------|----------------|----------|
| Control siRNA | Dharmacon | D-001810-01-05 | UGGUUUACAUGUCGACUAA |
| siCTNNA1 | Dharmacon | J-010505–06 | GAUGGUAUCUUGAAGUUGA |
| siCDC42 | Dharmacon | J-005057–07 | GAUGACCCCUCUACUAUUG |
| siCDH5 | Dharmacon | J-003641–07 | GAGCCCAGGUCAUUAUCAA |
| siFZD4 | Dharmacon | J-005503–06 | GAUCGAUUCUUCUAGGUUU |
| siFZD6 | Dharmacon | J-005505–07 | GAAGGAAGGAUUAGUCCAA |
| siFZD7 | Dharmacon | J-003671–11 | UGAUGUACUUUAAGGAGGA |
| siFZD8 | Dharmacon | J-003962–08 | UCACCGUGCCGCUGUGUAA |
| siROR1 | Dharmacon | J-003171–09 | UGACUUGUGUCGCGAUGAA |
| siROR2 | Dharmacon | D-003172–06 | GCAGGUGCCUCCUCAGAUG |
| siRYK | Dharmacon | J-003174–11 | GGUUUGUUGUGCAGUAAUA |
| siVCL | Dharmacon | J-009288–05 | UGAGAUAAUUCGUGUGUGUUA |
| siWNT11 | Dharmacon | L-009474-00-0005 | SMARTpool |
| siWNT5a | Dharmacon | L-003939-00-0005 | SMARTpool |

DOI: https://doi.org/10.7554/eLife.45853.022

## Mice and treatments

For the Cdc42 inhibition experiment, C57BL/6J mice were maintained at the Instituto de Medicina Molecular (iMM) under standard husbandry conditions and under national regulations. ML-141 (SML0407, Sigma, Germany) was injected twice (morning and evening) intraperitoneally (IP) (20 ml/g of 1 mg/mL solution) at postnatal day 5 (P5) before eyes were collected at P6.

*Vinculin* floxed mouse (*Zemljic-Harpf et al., 2007*) was obtained from Robert S. Ross *Pdgfb*-iCreERT2 (*Claxton et al., 2008*) to generate a new *Vinculin* fl/fl::Pdgfb-iCreERT2 mouse line. Mice were maintained at the Instituto de Medicina Molecular (iMM) under standard husbandry conditions and under national regulations. Animal procedures were performed under the DGAV project license 0421/000/000/2016. Tamoxifen (Sigma, Germany) was injected intraperitoneally (IP) (20 ml/g of 1 mg/mL solution) at postnatal day 1 (P1) and P3 before eyes were collected at P6.

**Table 2.** List of qPCR primers.

| Primer | Forward sequence | Reverse sequence |
|--------|------------------|------------------|
| CDC42 | TGACAGATTACGACCGCTGAGTT | GGAGTCTTTGGACAGTGGTGAG |
| CDH5 | TCTCCGCAATAGACAAGGACA | TGGTATGCTCCCGGTCAAAC |
| CTNNA1 | GGACCTGCTTTCGGAGTACATG | CTGAAACGTGGTCCATGACAGC |
| FZD4 | TTCACACCGCTCATCCAGTACG | ACGGGTTCACAGCGTCTCTTGA |
| FZD6 | GGCAGTGTATCTGAAAGTGCGC | GATGTGGAACCTTTGAGGCTGC |
| FZD7 | GTCTTCAGCGTGCTCTACACAG | ACGGCATAGCTCTTGCACGTCT |
| FZD8 | GCTCTACAACCGCGTCAAGACA | AAGGTGGACACGAAGCAGAGCA |
| GAPDH | GTCAAGGCTGAGAACGGGAA | TGGACTCCACGACGTACTCA |
| ROR1 | GAGGCAACCAAAACACGTCAGAG | GGCACACTCACCCAATTCTTCC |
| ROR2 | ACGTACCCTCGTGTAGTCC | CGATGACCAGTGGAATTGCG |
| RYK | CAGCAAGACCTGGTACACATGG | CAAGTCTCTGGAGAGGGCATTG |
| VCL | TGAGCAAGCACAGCGGTGGATT | TCGGTCACACTTGGCGAGAAGA |
| WNT5A | TACGAGAGTGCTCGCATCCTCA | TGTCTTCAGGCTACATGAGCCG |
| WNT11 | CAGTGTTGCGTCTGGTTCAGT | TGCTATGGCATCAAGTGGCT |

DOI: https://doi.org/10.7554/eLife.45853.024

For *Figure 1*, we re-used mouse retinas previously collected (*Franco et al., 2016*). For clarity, we transcribe the specificities of the breedings and experimental conditions. The following mouse strains were previously used: *Pdgfb*-iCreERT2 (*Claxton et al., 2008*); *Wnt5a* floxed (*Miyoshi et al., 2012*); *Wnt11* null (*Majumdar et al., 2003*). Mice were maintained at the London Research Institute under standard husbandry conditions. Tamoxifen (Sigma, Germany) was injected intraperitoneally (IP) (20 ml/g of 1 mg/mL solution) at postnatal day 2 (P2) before eyes were collected at P5 onwards. Animal procedures were performed in accordance with the Home Office Animal Act 1986 under the authority of project license PPL 80/2391.

## Immunofluorescence on mouse retinas

Eyes were collected at P6 and fixed with 2% PFA in PBS for 5 hr at 4°C, thereafter retinas were dissected in PBS. Blocking/permeabilisation was performed using Claudio's Blocking Buffer (CBB) (*Franco et al., 2013*), consisting of 1% FBS (Thermo Fisher Scientific), 3% BSA (Nzytech), 0.5% triton X100 (Sigma), 0.01% Na deoxycholate (Sigma), 0,02% Na Azide (Sigma) in PBS pH = 7.4 for 2 hr in a rocking platform. Primary antibodies (Anti-CD102 and Anti-Erg) were incubated at the desired concentration (see Key Resources Table) in 1:1 CBB:PBS at 4°C overnight in a rocking platform and afterwards washed $3 \times 60$ min in PBS-T. Then, retinas were incubated in 1:1 CBB:PBS solution containing the secondary fluorophore conjugated antibodies at 4°C overnight in the dark. Next, and due to the fact that we are using same species primary antibodies, retinas were incubated with AffiniPureF(ab')$_2$ fragments Donkey anti-rabbit IgG (see Key Resources Table) for 2 hr at RT, followed by 3 washes of 30 min in PBS-T. Retinas were fixed with 4%PFA in PBS at RT and blocked using CBB and primary antibody (Anti-GOLPH4) was incubated (see Key Resources Table) in 1:1 CBB:PBS at 4°C overnight in a rocking platform. Secondary antibody was done as previously described. Retinas were mounted on slides using Vectashield mounting medium (Vector Labs, H-1000, Burlingame, California, USA). For polarity quantification, a tile-scan spanning the sprouting front was acquired on a Zeiss Cell Observer Spinning Disk microscope, equipped with the Zen software with a Plan-Apochromat 40x/1.4 Oil DIC M27 objective.

## Culture of HUVECs

Human umbilical vein endothelial cells (HUVECs) were routinely cultured following the manufacturer's guidelines, in filter-cap T75 flasks Nunclon Δ surface treatment (VWR international, LLC) and cultured at 37°C and 5% $CO_2$ to ensure a stable environment for optimal cell growth. HUVECs (C2519A, Lonza) were cultured with complete medium EGM-2 Bulletkit (CC-3162, Lonza) supplemented with 1% penicillin/streptomycin (#15140122, Gibco). When passaging cells for experiments, cells were washed twice in sterile PBS (137 mM NaCl, 2.7 mM KCl, 4.3 mM $Na_2HPO_4$, 1.47 mM $KH_2PO_4$, pH7.4). Then, cells were incubated for 3–5 min in trypsin/EDTA (#15400054, Gibco) or in TrypLE Express (#12605–028, Gibco) at 37°C, 5% $CO_2$. When 95% of the cells detached, complete medium was added to each flask to inhibit the activity of the trypsin/EDTA or TrypLE Express and the cell suspension was transferred to a falcon tube. To maximize the amount of cells collected, all flasks were washed again with complete medium, which was added to the cell suspension gathered previously. HUVECs were then centrifuged at 115 g for 5 min at room temperature. The pellet was re-suspended in fresh complete medium. The cell concentration present in the suspension was determined using a Neubauer Chamber Cell Counting (Hirschmann EM Techcolor). All cells were then seeded on the desired culture vessels at 200.000–300.000 cells/mL and placed in the incubator. All experiments with HUVECs were performed between passages 3 and 6.

## siRNA transfection

In order to silence the expression of genes of interest, a set of ON-TARGET human siRNAs were purchased from Dharmacon (see *Table 1*). Briefly, HUVECs were seeded the day before the transfection to reach 60–70% confluence and were then transfected with 25 nM of siRNA using the DharmaFECT one reagent (Dharmacon, GE Healthcare) following the Dharmacon siRNA Transfection Protocol. 24 hr after transfection the culture medium was replaced by fresh complete medium and cells were kept under culture conditions up until 72 hr post-transfection and then processed for further experiments. siRNA efficiencies were measured by qPCR and by WB when antibodies were available (*Figure 11*).

## RNA extraction and quantitative Real-Time PCR

RNA extraction was performed from HUVECs seeded on 12-well plates using the RNeasy Mini Kit (Qiagen) and the GeneJet RNA Purification Kit (Thermo Scientific) as described by the manufacturer's protocol. RNA concentration was quantified using NanoDrop 1000 (Thermo Scientific) and adjusted equally, followed by DNase I digestion (Thermo Scientific) and cDNA synthesis (Superscript IV First-Strand Synthesis System, Invitrogen). cDNA samples were then diluted in RNAse/DNAse-free water for the subsequent quantitative real-time PCR (RT-qPCR) reactions. RT-qPCR was performed using a 7500 Fast Real-Time PCR System (Applied Biosystems) with Power SYBR Green PCR Master Mix (Applied Biosystems) following the standard program of the system previously mentioned. For each reaction, 5 μL of cDNA was combined with 10 μL of Power SYBR Green PCR Master Mix, 4.5 μL of RNAse/DNAse free water and 0.5 μL of 4 μM primers pool (Forward +Reverse) (see *Table 2*) in a MicroAmp Fast Optical 96-well Reaction Plate (Applied Biosystems). The expression levels of each

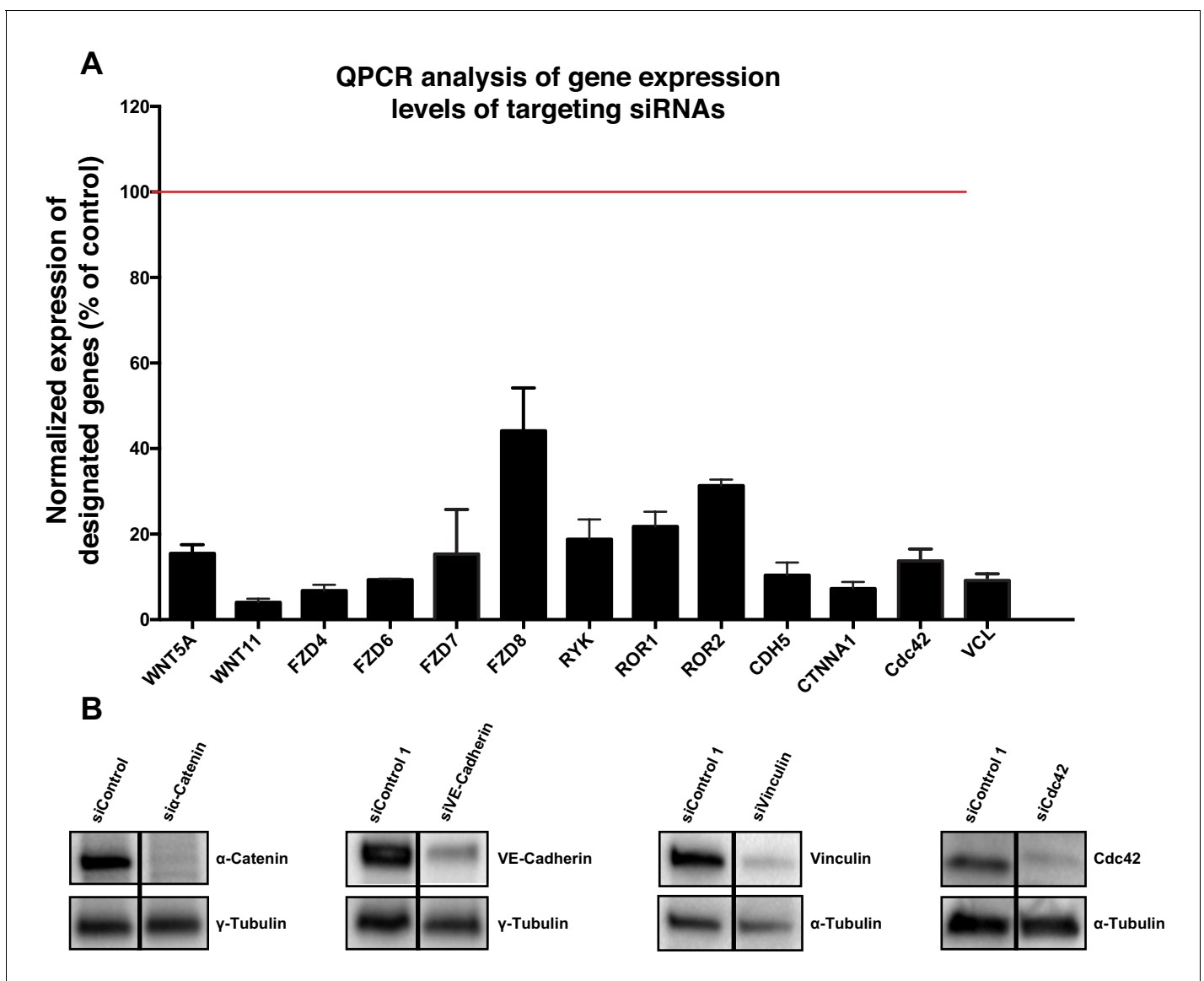

**Figure 11.** Validation of specificity of siRNAs used in this study. (**A**) Quantification of mRNA expression levels by qPCR showing the knockdown efficiencies of siRNAs against CDC42, CDH5, CTNNA1, FZD4, FZD6, FZD7, FZD8, ROR1, ROR2, RYK, VCL, WNT5a and WNT11. Data are mean ± SD, gene expression levels were normalized to GAPDH. (**B**) Western blot showing siRNA knockdown efficiency for α-Catenin (n = 2), VE-cadherin (n = 2), vinculin (n = 1) and Cdc42 (n = 1).

DOI: https://doi.org/10.7554/eLife.45853.023

sample duplicate were then normalized to GAPDH and the $2^{-\Delta\Delta T}$ method was used to calculate relative alterations in gene expression (*Figure 11*).

## Protein extraction and western blotting

Protein extraction was performed from HUVECs seeded on 6-well plates which were lysed in 120 μL of RIPA buffer (50 mM Tris/HCl pH7.5, 1% NP-40, 150 mM NaCl, 0.5% sodium deoxycholate, 0.1% SDS in $H_2O$) supplemented with phosphatase and proteinase inhibitors cocktail (1:100, #10085973 Fischer Scientific). Adherent cells were then detached from the plate with a cell scrapper and the cell lysates were gathered and transferred into an ice cold eppendorf tube. The cell lysates were then centrifuged at maximum speed for 10 min at 4°C and the supernatants collected into a new eppendorf tube. Protein concentration was quantified using the BCA protein assay kit (Pierce) following the guidelines recommended by the manufacturer. The Multimode microplate reader, Infinite M200 (Tecan), was used for spectrophotometric measurement of protein with the i-control software. For Western Blotting protein samples were normalized up to 25 μL and combined with a mixture of 2x Laemmli Sample Buffer (#161–0747, Bio-rad Laboratories) with 450 mM DTT (D0632, Sigma-Aldrich) and incubated at 70°C in a Dry Block Thermostat (Grant Instruments, Ltd) for 10 min (or 95°C for 5 min). Protein samples were loaded and separated on a 4–15% Mini-PROTEAN TGX Gel (#456–1084, BioRad) along with 5 μL of protein ladder (Full-Range RPN800E, GE Healthcare Rainbow Molecular Weight Markers), first at 50V for 5 min and then at 100–130V for 1–2 hr in SDS-PAGE running buffer (10x SDS-PAGE: 250 mM Tris, 1.92M Glycine, 1% SDS, pH8.3).

Gels were then transferred either onto a nitrocellulose membrane (iBlot Transfer Stack Regular/Mini size, #IB3010-01/−02, Invitrogen) with iBlot Dry Blotting System (Invitrogen) for 4–7 min; or onto a Polyvinylidene Difluoride (PVDF) membrane (#IPVH00010, Merck Milipore) with Mini Trans-Blot Electrophoretic Transfer Cell (Biorad) following the manufacturer's guidelines. After transfer, blotted membranes were incubated in Ponceau Red to assess transfer quality, and then washed in TBS-T (50 mM Tris/HCl, 150 mM NaCl, 0.1% Tween-20, pH7.5). Then, membranes were incubated in blocking buffer containing 3% BSA (Bovine Serum Albumin, MB04602, Nzytech) in TBS-T for 1 hr at RT, followed by an overnight incubation at 4°C with the primary antibodies diluted in the same blocking buffer (see Key Resources Table).

On the following day membranes were washed three times in TBS-T and incubated in blocking buffer containing the secondary horseradish peroxidase (HRP) conjugated antibodies for 1 hr at RT (see Key Resources Table).

Before revelation membranes were washed again three times in TBS-T for 5 min and then incubated in ECL Western Blotting Detection Reagent (RPN2209, GE Healthcare) following the manufacturer's protocol.

Protein bands were visualized in Chemidoc XRS + and relative protein quantities were measured using the Image Lab software, both from Bio-Rad Laboratories. All results were normalized to tubulin levels.

## Pulldown of active GTP-bound Cdc42

Active Cdc42 pulldown was performed from HUVECs cultured in 10 cm plates non-stimulated or stimulated with recombinant human Wnt5a protein (645-WN, R and D Systems, 200 ng/mL) for 15 min using the Cdc42 Pull-down Activation Assay Biochem Kit (Cytoskeleton) as described by the manufacturer's protocol. Briefly, after stimulation, cells were washed with ice cold PBS, scrapped and lysed in lysis buffer containing protease and phosphatase inhibitors. After lysate clarification, inputs from all the samples were gathered and the remaining lysate was used for the pulldown reaction. 10 μg of PAK-PBD beads were added to equivalent protein amounts of cell lysates (300 μg) for each condition. The mixture was then incubated for 1 hr at 4°C with gentle rotation. After the pulldown reaction, beads were washed three times in washing buffer and the bound protein complexes were eluted in sample buffer with DTT by placing the beads for 5 min at 95°C. Samples were then blotted on SDS-PAGE following standard protocols.

## Immunoprecipitation

VE-cadherin and vinculin immunoprecipitation was performed from HUVECs cultured in 10 cm plates. After the scratch-wound assay, cells were incubated with PBS supplemented with 1 mM

CaCl$_2$ and 0.5 mM DSP (#22585, Thermo Scientific) for 20 min at RT. Afterwards they were washed twice with ice cold PBS and then four times with ice cold quenching buffer (10 mM Tris/HCl, pH 7.5, in PBS). Then, cells were scrapped and lysed in lysis buffer (25 mM Tris/HCl, pH 7.5, 1% NP-40, 1% deoxycholic acid, 150 mM NaCl) containing protease and phosphatase inhibitors. Cell lysates were centrifuged at 16,100 g for 10 min at 4°C and the pellet digested in SDS IP buffer (15 mM Tris/HCl, pH 7.5, 5 mM EDTA, 2.5 mM EGTA, 1% SDS). Samples were then incubated for 10 min at 100°C and diluted in lysis buffer. At this point, inputs from all the samples were gathered and the remaining lysate was used for immunoprecipitation. Pre-washed Pierce G-protein agarose beads (#22851, Thermo Scientific) were added to equivalent protein amounts of cell lysates (100–200 µg) for each condition, containing either 2 µg of anti-vinculin (V9264, Sigma-Aldrich) or anti-VE-cadherin (sc-9989, Santa Cruz Biotechnologies) antibody. The mixture was then incubated overnight at 4°C with gentle rotation. After immunoprecipitation, beads were washed four times in ice cold lysis buffer and the bound protein complexes were eluted in sample buffer with DTT by placing the beads for 10 min at 100°C. Samples were then blotted on SDS-PAGE following standard protocols.

## Immunofluorescence

For immunofluorescence of in vitro cultured HUVECs, cells were seeded on 24-well plates with glass coverslips, or in 8-well Ibidi slides (80826, Ibidi) previously coated with 0.2% Gelatin in sterile water (G1393, Sigma-Aldrich) or with Fibronectin in PBS (F1141, Sigma-Aldrich), respectively. After the scratch-wound assay (described above), HUVECs were fixed in 1% Paraformaldehyde (PFA) supplemented with 1M MgCl$_2$ and 1M CaCl$_2$ (1 µL/2 mL) in PBS for 30 min at RT. Cells were then washed with PBS to remove the remaining PFA and the immunostaining protocol initiated. When the PBS was removed, HUVECs were blocked and permeabilized with blocking solution containing 3% BSA in PBS-T (PBS with 0.1% Triton X-100) for 30 min at RT. Then cells were incubated for 2 hr at RT with the primary antibodies diluted in the blocking solution (see Key Resources Table) and washed 3 × 15 min washes in PBS-T. Afterwards, cells were incubated in blocking solution containing the secondary fluorophore conjugated antibodies for 1 hr at RT in the dark, followed again by 3 washes of 15 min in PBS-T. Finally, HUVECs were incubated with 1x DAPI (D1306, Molecular Probes by Life Technologies) diluted in PBS-T for 5 min in the dark. Coverslips were then mounted on microscopy glass slides using Mowiol DABCO (Sigma-Aldrich), while for the 8-well Ibidi slides 50 µL of Mowiol DABCO was added to each well. To quantify co-localization of junctional molecules, high-resolution Z-stack images at multiple positions on the scratch front were acquired on a confocal Laser Point-Scanning Microscope 880 (Zeiss) equipped with the Zen black software with a Plan Apochromat 63x NA 1.40 oil DIC M27 objective. For polarity quantification, a tile-scan spanning the entire region of the scratch was acquired on a motorized inverted widefield fluorescence microscope, Zeiss Axiovert 200M (Carl Zeiss) equipped with the Metamorph software with an EC Plan-NeoFluar 40x NA 0.75 dry objective.

## Immunostaining co-localization analysis

For co-localization analysis, high-resolution Z-stack confocal images of HUVECs stained for junctional proteins (VE-Cadherin, Vinculin, α-catenin, β-catenin and p120-catenin) were imported and analyzed in MATLAB using a custom written code (*Source code 1*). An object-based co-localization approach was used. Briefly, each channel was segmented and a binary mask was generated. The masks were combined and the fraction of pixels with overlapping signals was quantified.

## Calcium switch assay

Confluent HUVECs seeded on 24-well plates were subjected to the scratch-wound assay and then incubated for 15 min in Ca2 +free HBSS, followed by DMEM (#41966–029, Gibco) supplemented with 1% penicillin/streptomycin (#15140122, Gibco), 10% fetal bovine serum (FBS) (#10500–064, Gibco) and 2 mM Ca$^{2+}$ from 1 up to 30 min at 37°C, 5% CO$_2$. Afterwards, cells were immediately fixed in 1% Paraformaldehyde (PFA) supplemented with 1M MgCl$_2$ and 1M CaCl$_2$ (1 µL/2 mL) and processed for immunostaining.

## Proximity ligation assay (PLA)

Confluent HUVECs seeded on 24-well plates were subjected to the scratch-wound assay and then processed for PLA using the Duolink In Situ Red Mouse/Rabbit Starter Kit (DUO92101-1KT, Sigma-Aldrich) as described by the manufacturer's protocol. To probe interactions between vinculin and VE-cadherin, cells were incubated with an anti-vinculin antibody raised in rabbit (V4139, Sigma-Aldrich) and an anti-VE-cadherin antibody raised in mouse (sc-9989, Santa Cruz Biotechnologies). In parallel, cells were also incubated with an anti-VE-cadherin antibody raised in goat (AF938, R and D Systems) and subsequently with an anti-Goat fluorescent-conjugated secondary antibody (A21447, Thermo Fisher Scientific) to label adherens junctions. To quantify co-localization of PLA signal at adherens junctions, high-resolution Z-stack images at multiple positions on the wound edge were acquired on a confocal Laser Point-Scanning Microscope 880 (Zeiss) equipped with the Zen black software with a Plan Apochromat 63x NA 1.40 oil DIC M27 objective. Briefly, PLA dots were quantified only at adherens junctions, using a similar approach to the co-localization studies described in the section 'Immunostaining co-localization analysis', using the VE-cadherin immunofluorescence staining to detect overlapping pixels between junctions and PLA signals.

## Viral production and transduction

Replication-incompetent lentiviruses were produced by transient transfection of HEK293T of pLX303 lentiviral expression vector co-transfected with the viral packaging vector Δ8.9 and the viral envelope vector VSVG. Medium was replaced with fresh culture medium 5–6 hr post transfection. 48 hr after medium replacement, lentiviral particles were concentrated from supernatant by ultracentrifugation at 90000 g for 1h30 and re-suspended in 0.1% BSA PBS. Seeded HUVECs were transduced 24 hr post-transfection with varying concentrations of lentiviral plasmids containing VE-Cad-TS, VE-Cad-TL, Cdc42-2G, Vinculin-Full-Length-GFP, Vinculin-T12-mutant-GFP and αCat-Vinc-HA fusion protein sequences (see Key Resources Table). 24 hr after viral transduction the culture medium was replaced by fresh complete medium and cells were kept in culture conditions up until 48 hr post-transduction and then processed for immunofluorescence or imaging. In the analysis, we used a mix population containing transduced and non-transduced cells, selecting areas where high transduction efficiencies were observed.

## Scratch-wound assay and drug treatments

To assess functional collective cell behavior properties (*i.e.*, polarity and migration), as well as morphological features of in vitro cultured HUVECs, we used the scratch-wound assay. The wound was created by scratching the surface of a well-plate or a microscopy glass slide containing a monolayer of adherent HUVECs with a 200 μL pipette tip. The culture medium was then replaced by fresh complete medium and HUVECs were allowed to migrate under optimal physiological conditions. When appropriate, drugs of interest were added to the medium. (see Key Resources Table). Cells migrated for 5 hr after the wound, were fixed and then stained for immunofluorescence experiments. For live imaging experiments HUVECs migration was followed up to 16 hr. Imaging was performed using a Zeiss Cell Observer SD (Carl Zeiss) equipped with an EC Plan-Neofluar 10x NA 0.3 PH1. To track individual cells within the monolayer more efficiently using the cell nuclei as reference, HUVECs were incubated in 1x Hoechst for 15 min at 37°C before the onset of the time lapse. Images of the scratch front were acquired at multiple positions every 10 min. Analysis of migration, including wound closure, cell speed and straightness was performed using FIJI TrackMate plug in and the Chemotaxis and Migration Tool (free software from Ibidi).

## Particle image velocimetry (PIV) analysis

The velocity field of the moving cell sheet was calculated in Matlab using cell image velocimetry (CIV) (*Milde et al., 2012*) software. Interrogation windows were set to $64 \times 64$ pxls with a 50% overlap. Velocity spatial correlation was calculated in Matlab using the x-component of the velocity as in *Petitjean et al. (2010)*. Correlation length was determined from exponential fitting of correlation curves.

## Atomic force microscopy (AFM)

HUVECs were re-plated onto 35 mm Petri dishes (TPP) 48 hr post-transfection from 6-well plates (on a ratio of 1 6-well plate to 2 35 mm Petri dishes per condition) to attain a confluence of 60–70%. On the following day, 1 hr before starting the cell-cell adhesion measurements, the culture medium was replaced by PBS in one of the 35 mm Petri dish replicates, to ensure cell detachment. 5 min before the experiment, the culture medium of the other 35 mm Petri dish replicate was replaced with serum-free culture medium. An atomic force microscope NanoWizard II (JPK Instruments, Berlin, Germany) mounted on the top of an Axiovert 200 inverted microscope (Carl Zeiss, Jena, Germany) was used for the cell-cell adhesion measurements. For these experiments, tipless arrow TL1 cantilevers (Nanoworld, Neuchatel, Switzerland) were used, with a nominal spring constant of 0.03 N m$^{-1}$, as described previously (*Ribeiro et al., 2016*). Cantilevers were cleaned for 15 min with UV light and coated with poly-D-lysine (50 µg ml$^{-1}$) for at least 30 min. Cantilevers were stored in poly-D-lysine solution until use.

After that, a set of adherent cells from the other Petri dish were selected to perform the cell-cell adhesion measurements, composed of 5 force-distance curves performed on each cell, with a cell-cell contact time of 5 s and a 5 s pause between them. Cell–cell contact was established with an applied force of 300 pN, at a constant height and in closed-loop mode. The AFM tip resonant frequency was maintained at 2 Hz, with a z-range displacement of 50 µm. For the internal negative controls, we used 4 mM EGTA, a Ca$^{2+}$ chelating agent that is able to sequestrate calcium ions from cadherins and render them inactive and unresponsive to force transmission. EGTA was added to the serum-free culture medium of the Petri dish containing the adherent cells at the time of the recordings.

## Analysis of tension sensors FRET measurements

FRET images were obtained using a confocal Laser Point-Scanning Microscope 880 (Zeiss) equipped with a Plan-Apochromat 63x, NA 1.40, oil immersion, DIC M27 objective and an argon laser featuring 405, 458 and 514nm laser lines. For FRET experiments, it was used a MBS 458/514 beam splitter in combination with the following filters: mTFP1 GaAsP, band-pass 461–520; Venus/FRET, band-pass 525–575. Acceptor photobleaching experiments were analyzed using a custom written Matlab script (*Source code 2*). A Gaussian filter with standard deviation of 0.75 was applied to the images before analysis. The intensity in the region of interest was measured before and after bleaching. FRET efficiency was calculated as $EF = \frac{I_{post} - I_{pre}}{I_{post}}$ where I$_{post}$ and I$_{pre}$ are the intensity of the donor channel after and before bleaching respectively.

## Analysis of CDC42 biosensor data

The CDC42-2G FRET biosensor activity was obtained using a widefield fluorescence microscope Axio Observer (Zeiss) equipped with a Plan-Apochromat 63x, NA 1.40, oil immersion, DIC M27 objective. For ratiometric FRET experiments we used the following excitations and emission filters: ET436/20 and ET480/40 for ECFP; ET500/20 and ET535/30 for EYFP (Chroma Technology Corp) and the images for each condition were acquired during 5 min with 1 s time interval. FRET experiments were performed as described by Louis Hodgson. Analysis of ratiometric FRET biosensor was performed in Matlab and the preprocessing was performed using the Biosensor Processing 2.1 software package from the Danuser lab (*Hodgson et al., 2010*).

The resulting images showing the localized activation of CDC42 were further processed to retrieve quantitative information from such maps. Briefly, junctional or free-edges regions were selected from each time-lapse image and the differential of the intensity vs time traces was calculated. For each image a region where no activation was detected was also selected to determine the level of background signal. The local maxima for each curve above background level were determined. Maxima found within three frames from each other were assumed to correspond to the same activation event.

## Polarity index calculation

To quantify cell polarity, tile-scan images of HUVECs stained with Golgi (Golph4) and nuclear (DAPI) markers were processed on Adobe Photoshop to separate leader cells, identified as the first row of cell directly in contact with the scratch, from follower cells, comprising the second to fourth rows of

cells away from the scratch. Afterwards, each set of images was imported and analyzed in MATLAB using a modified version of a polarity analysis script kindly provided by Anne-Clémence Vion and Holger Gerhardt. Briefly, after segmenting each channel corresponding to the Golgi and nuclear staining, the centroid of each organelle was determined and a vector connecting the center of the nucleus to the center of its corresponding Golgi apparatus was drawn. The Golgi-nucleus assignment was done automatically minimizing the distance between all the possible couples. The polarity of each cell was defined as the angle between the vector and the scratch line. An angular histogram showing the angle distribution was then generated. Circular statistic was performed using the Circular Statistic Toolbox.

To test for circular uniformity, we applied the polarity index (PI), calculated as the length of mean resultant vector for a given angular distribution (*Figure 1D*).

$$PI = \sqrt{\left(\frac{1}{N}\sum_1^N \cos\alpha\right)^2 + \left(\frac{1}{N}\sum_1^N \sin\alpha\right)^2}$$

PI corresponds to length of the mean resultant vector, previously described in *Berens (2009)*. The PI varies between 0 and 1. The closer to one the more the data are concentrated around the mean direction, while values close to 0 corresponds to random distribution. PI indicates the collective orientation strength of the cell monolayer. Box plots were generated by using every single PI calculated for images of each biological replica, which show the biological variability of the system. This data is used to calculate the significance of differences between experimental conditions.

To obtain a global description of a given experimental condition, we pooled together all the different biological replicates in one single file and calculate a global PI and mean angle (angular histograms and values below in the main text). This representation provides information on the general distribution of polarities in all experiments, and provides a mean angle of polarity, which is important to understand directionality.

To calculate the PI as a function of distance, each image was divided starting from the wound edge in slices 50 μm apart. The cell polarity within each slice was extracted and represented as angular histogram and the corresponding PI was calculated. For *Figure 1E,a* global polarity index was calculated merging together the results from different images from the same experimental conditions. N = 9 for siControl, N = 10 for siWNT5a and N = 4 for siCtnna1.

## Statistical analysis

All statistical analysis was performed using GraphPad Prism 7 and Matlab (Mathworks). Statistical details of experiments are reported in the figures and figure legends. Sample size is reported in the figure legends and no statistical test was used to determine sample size. The biological replicate is defined as the number of cells, images, animals, as stated in the figure legends. No inclusion/exclusion or randomization criteria were used and all analyzed samples are included. Comparisons between two experimental groups were analyzed with unpaired parametric t test, while multiple comparisons between more than two experimental groups were assessed with one-way ANOVA. We considered a result significant when p<0.05. Box plots for polarity indexes represent min to max, central line represents mean.

## Acknowledgements

The authors thank M Schwartz (Yale University, New Haven) for plasmids; RS Ross (University of California, San Diego) for providing the Vinculin floxed mouse; M Takeichi (RIKEN, Kobe) for a18 antibody; AC Vion and H Gerhardt (Max-Delbruck Center, Berlin) for providing the basis of our Matlab script for automated polarity analysis; and M Nakayama (MPI Bad Nauheim), J Barata, D Henrique, and E Gomes (iMM, Lisbon) for discussions. Research was supported by European Research Council starting grant (679368), the H2020-Twinning grant (692322), the Fundação para a Ciência e a Tecnologia funding (grants: IF/00412/2012; EXPL-BEX-BCM-2258–2013; PRECISE-LISBOA-01–0145-FEDER-016394; UID/BIM/50005/2019, a project funded by Fundação para a Ciência e a Tecnologia (FCT)/ Ministério da Ciência,Tecnologia e Ensino Superior (MCTES) through Fundos do Orçamento de Estado; and a grant from the Fondation Leducq (17CVD03); and personal fellowships: BD/52224/2013 to JRC, BD/105856/2014 to PB, and BD/128375/2017 to CF) and LISBOA-01–0145-FEDER-

007391, project cofunded by FEDER, through POR Lisboa 2020 - Programa Operacional Regional de Lisboa, PORTUGAL 2020, and Fundação para a Ciência e a Tecnologia. Authors declare no competing interests.

## Additional information

### Funding

| Funder | Grant reference number | Author |
| --- | --- | --- |
| H2020 European Research Council | 679368 | Claudio A Franco |
| Fundação para a Ciência e a Tecnologia | IF/00412/2012/CP0163/CT0007 | Claudio A Franco |
| Fondation Leducq | 17CVD03 | Claudio A Franco |
| H2020 Spreading Excellence and Widening Participation | 692322 | Claudio A Franco |

The funders had no role in study design, data collection and interpretation, or the decision to submit the work for publication.

### Author contributions

Joana R Carvalho, Conceptualization, Data curation, Formal analysis, Investigation, Methodology, Writing—original draft, Writing—review and editing; Isabela C Fortunato, Data curation, Investigation, Methodology, Writing—review and editing; Catarina G Fonseca, Conceptualization, Data curation, Investigation, Methodology, Writing—review and editing; Anna Pezzarossa, Conceptualization, Software, Formal analysis, Supervision, Investigation, Methodology, Writing—review and editing; Pedro Barbacena, Maria A Dominguez-Cejudo, Formal analysis, Investigation, Writing—review and editing; Francisca F Vasconcelos, Validation, Investigation, Methodology, Writing—review and editing; Nuno C Santos, Supervision, Methodology, Writing—review and editing; Filomena A Carvalho, Data curation, Validation, Investigation, Methodology, Writing—review and editing; Claudio A Franco, Conceptualization, Data curation, Software, Supervision, Funding acquisition, Validation, Investigation, Visualization, Methodology, Writing—original draft, Project administration, Writing—review and editing

### Author ORCIDs

Francisca F Vasconcelos (iD) http://orcid.org/0000-0003-0601-5513
Claudio A Franco (iD) https://orcid.org/0000-0002-2861-3883

### Ethics

Animal experimentation: Mice were maintained at the Instituto de Medicina Molecular (iMM) under standard husbandry conditions and under national regulations, under the license AWB_2015_11_CAF_Polaridade.

### Decision letter and Author response

Decision letter https://doi.org/10.7554/eLife.45853.030
Author response https://doi.org/10.7554/eLife.45853.031

## Additional files

### Supplementary files

• Source code 1. Colocalization matlab source code.
DOI: https://doi.org/10.7554/eLife.45853.025

• Source code 2. Acceptor photobleaching matlab source code.
DOI: https://doi.org/10.7554/eLife.45853.026

• Transparent reporting form
DOI: https://doi.org/10.7554/eLife.45853.027

**Data availability**

All data generated or analyzed during this study are included in the manuscript and supporting files.

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
