## [Decision Letter]

Thank you for submitting your article "Non-canonical Wnt signaling regulates junctional mechanocoupling during angiogenic collective cell migration" for consideration by *eLife*. Your article has been reviewed by three peer reviewers, and the evaluation has been overseen by a Reviewing Editor and Jonathan Cooper as the Senior Editor. The reviewers have opted to remain anonymous.

The reviewers have discussed the reviews with one another and the Reviewing Editor has drafted this decision to help you prepare a revised submission.

The comments are relatively favorable and constructive. However, the authors are required to address the following.

1) Please carefully address the major comments 1 and 2 of reviewer 1 with additional experiments and/or alternative analyses. In particular, it seems necessary to validate the usefulness of polarity index (PI) *in vivo* in retinal angiogenesis, regarding whether or not the geometric effects of vascular pattern could be neglected. In addition, please clarify whether formula in Figure 1D (subsection “Non-canonical Wnt signaling is required for the coordination of collective cell polarity *in vitro* and *in vivo*”, second paragraph) was developed by the authors or was modified from a previous study.

2) Please address the other points of reviewer 1 and 3 point-by-point in a data-driven manner or with further analyses.

I believe the authors are capable of addressing most of the comments, but please provide the reasons for not implementing the suggested changes where necessary.

*Reviewer #1:*

In this report, the authors propose a biomechanical mechanism regulated by non-canonical Wnt signaling in angiogenic collective movements of ECs. They first tried to develop a simple assay system, which enables evaluation of endothelial collective movement in *in vitro* wound healing and *in vivo* murine retinal angiogenesis. Using the key assay system, they identified that non-canonical Wnt signaling coordinates collective cell polarity by regulating mechanical tension via vinculin stability control at EC adherent junctions. They further showed that the junctional mechanocoupling is fine-tuned by non-canonical Wnt signaling through the ROR2-Cdc42 signaling axis. The findings are interesting and could provide novel and important mechanistic information in the field of angiogenic morphogenesis, and on the whole the experimental designs seem well organized. However, the following issues need to be addressed.

1) To more simply quantify the extent of coordination between cells at population levels compared to PIV analysis-based method, the authors generated a method using a polarity index (PI), which is developed from axial polarity index. The reviewer agrees that the index reflects well the extent of EC coordination in scratch wound assay *in vitro*. However, the index does not seem to reflect the coordination status correctly and sufficiently if they adopted the same algorithm for analysis in retinal angiogenesis *in vivo* because of the following reason. In case of scratch wound assay, EC theoretically can move toward all directions without specific spatial limitations, although the moving direction tended to be biased toward wound area. In the case of retinal angiogenesis, moving direction is spatially limited or biased by the vascular structure and network pattern, in which ECs move. In other words, vascular geometry pattern can potently affect the PI. For example, increasing the number of vascular branches in parallel with spouting front edge in the analyzed area, PI is predicted to become smaller. Actually, PI value in murine retinal angiogenesis was smaller than in those in *in vitro* scratch wound assay as seen in Figure 1N. Because, in the logical flow of this manuscript, PI serves as a key assay parameter even *in vivo*, they need to more carefully evaluate the validity of PI also *in vivo*. In this sense, they may wish to consider that leading front edge is set in each branch to calculate PI instead of sprouting front edge as shown in Figure 1L.

2) In some cell biology experiments, explanation of experimental protocols and data presentation seem insufficient in the text including sections of Materials and methods, Results and figure legends, which sometimes caused the reviewer difficulties in understanding the data completely. Specifically, for rescue or overexpression experiments using Lenti virus vector in scratch wound assay (Figure 1H and J, Figure 7 and Figure 8), did the authors use only transduced cells after cell sorting or not? If not, what percentage of cells used were transduced in each experiment? Transduction efficiency can potentially affect the experimental result. If they assayed in mosaic condition mixed with no transduced cells, they may wish to compare the effect between transduced and non-transduced cells in same dish. Also, biochemical data showing the levels of exogenous proteins expressed using overexpression vector were omitted in all rescue experiments.

Specific points:

1) The reviewer could not understand how "the correlation length" was calculated even though the reviewer referred to Ng et al., 2012 and Petitjean et al., 2010. More detailed information needs to be described anywhere in the text.

2) Do the authors have any explanations as to why the coordination of leader cells is significantly affected by Wnt5A KD?

3) Biochemical data and the quantification shown in Figure 4A and B are confusing to the reviewer. β-catenin levels seem higher in control group than in Wit5A KD group in the representative data of western blotting in Figure 4A, while the mean level is rather lower in the control in the quantification data of Figure 4B although there is no statistical significance. Which is correct? Also, vinculin levels are significantly higher in Wint5a KD group in Figure 4B, but they describe that there were not any differences there in the text (subsection “Non-canonical Wnt signaling regulates vinculin stability at adherens junctions to reinforce junctional mechanocoupling”).

4) Is the overall wound closure impaired in vinculin KD ECs concomitant with the decrease of PI?

5) The authors should change the description of Figure 8D to Figure 8E (subsection “Constitutively active vinculin is sufficient to rescue collective behavior defects in Wnt5a-deficient endothelial cells”).

6) Figure 9H needs to be quantified similarly.

7) There are no headings in each figure legend throughout the manuscript.

8) The authors need to state how they considered statistical significance in the text.

9) The authors need to change the description to the correct one in the figure legend for Figure 1C.

*Reviewer #2:*

In this paper, Carvalho et al. investigate mechanisms regulating collective cell migration of endothelial cells during angiogenic sprouting. The authors report that the non-canonical Wnt ligand Wnt5a plays a crucial role in regulating the collective cell behavior of endothelial cells. Using various *in vitro* and *in vivo* approaches, they find that Wnt5a, through its receptor ROR2, activates Cdc42 to support the coupling between adherens junctions and the actin cytoskeleton. This mechanism involves the stabilization of the vinculin-α-catenin interaction, which is important for force transmission between individual endothelial cells.

The mechanisms that control collective endothelial migration are poorly understood. Here Carvalho and colleagues use elegant methods to demonstrate that non-canonical Wnt signaling plays a vital role in regulating the collective behavior of endothelial cells during angiogenic sprouting.

Overall, the analyses are thorough and of good quality, and the results nicely presented. The findings are relevant to the field, and there will also be an interest in the described methods to study the collective behavior of endothelial cells.

*Reviewer #3:*

Carvalho et al. report that non-canonical Wnt5a signaling reinforces coupling between VE-cadherin and the actin cytoskeleton in response to mechanical stimuli. They investigated this function in the context of the collective endothelial cell migration such as occurs during angiogenesis. The authors show that Wnt5a is cell autonomously required for endothelial cells to coordinate their collective migration both in an *in vitro* scratch wound assay and during angiogenesis *in vivo*. Cells lacking Wnt5a did not form high-tension adherens junctions. VE-cadherin was under less tension based on a FRET tension-sensor, and endothelial cells formed weaker cell-cell interactions based on atomic force microscopy measurements. This compromised the ability of cells to coordinate their polarity and migration. Mechanistically, the authors report that the non-canonical receptor ROR2 is necessary for transducing the Wnt5a signal via CDC42 activation. The authors exclude other non-canonical receptors and other signaling pathways downstream of ROR2. CDC42 promotes co-localisation of vinculin with VE-cadherin, connecting it to the actin cytoskeleton.

The same group have previously reported that non-canonical Wnt signaling regulates the sensitivity of ECs to blood flow by regulating polarity. In this report, they show that non-canonical Wnt regulates EC polarity in response to junctional pulling, providing new insight into how non-canonical Wnt can regulate mechanosensing by endothelial cells during angiogenesis.

Overall the manuscript is well written, the data are comprehensive and generally support the conclusions. There are a few points that would strengthen it further.

1) The claim for reduced association between actin stress fibers and VE-cadherin is not clear from the data shown (Figure 2B and Figure 9H). Some form of quantification is required to support this.

2) Vinculin is shown to co-localize with VE-cadherin at junctions by immunofluorescence imaging (Figure 4C) yet by PLA, sites of co-localisation appears to be scattered across the cell at random and not enriched at junctions. What are the implications of this distribution pattern for how vinculin and VE-cadherin regulate junction tension?

3) Was the co-immunoprecipitation in Figure 4G performed on cells that had been wounded? If not, what is the source of tension that promoted VE-cad and vinculin co-association and would this be stronger in wounded monolayers?

4) Retina vessels in vinculin ECKO mice (Figure 6C and D) show some phenotypic overlap with Wnt5a/Wnt11 mutants. This includes reduced vessel density which in Wnt5a/Wnt11 mice is due to increased vessel regression. Do the Vincluin ECKO mice have increased vessel regression? The authors should quantify this and discuss the results in the context of the junction mechanocoupling mechanism they describe.

---

## [Author Response]

Reviewer #1:[…] 1) To more simply quantify the extent of coordination between cells at population levels compared to PIV analysis-based method, the authors generated a method using a polarity index (PI), which is developed from axial polarity index. The reviewer agrees that the index reflects well the extent of EC coordination in scratch wound assay *in vitro*. However, the index does not seem to reflect the coordination status correctly and sufficiently if they adopted same algorithm for analysis in retinal angiogenesis *in vivo* because of the following reason. In the case of scratch wound assay, EC theoretically can move toward all directions without specific spatial limitations, although the moving direction tended to be biased toward wound area. In case of retinal angiogenesis, moving direction is spatially limited or biased by the vascular structure and network pattern, in which ECs move. In other words, vascular geometry pattern can potently affect the PI. For example, increasing the number of vascular branches in parallel with spouting front edge in the analyzed area, PI is predicted to become smaller. Actually, PI value in murine retinal angiogenesis was smaller than in those in *in vitro* scratch wound assay as seen in Figure 1N. Because, in the logical flow of this manuscript, PI serves as a key assay parameter even *in vivo*, they need to more carefully evaluate the validity of PI also *in vivo*. In this sense, they may wish to consider that leading front edge is set in each branch to calculate PI instead of sprouting front edge as shown in Figure 1L.

We thank the reviewer for this very relevant and insightful comment. Indeed, as the reviewer points out, PIs for analyses *in vivo* are generally smaller than PI values from *in vitro* experiments. These differences can be derived from the complexity of the *in vivo* microenvironment when compare to the relatively simple organisation of the *in vitro* wounded monolayers. Thus, we agree with the reviewer that geometry of vessel segments are imposing constraints into the freedom of endothelial cells to migrate/polarise in any direction. Therefore, those constraints can restrict the amplitude of PI.

We performed analysis of branch orientation in three different experimental conditions from an ongoing work: intraocular injections of PBS, VEGF and sFLT1. Retinas were treated for 36h. Three retinas per condition were analysed. Although the phenotype is different between the three conditions, the branch directionality is preserved in the sprouting front (Author response image 1). Kolmogorov-Smirnov test on distribution of the angles formed by the branches with the sprouting front edge showed confirmed that the underlying distribution was the same in the three different conditions.

Inversely, PI values were significantly different between the three conditions. In the case of sFlt1, the cells preferentially oriented toward the optic nerve, thus we attributed to these retinas a negative value of the PI, as cells are not orientated by VEGF gradients but rather by blood flow.

Based on these results, we believe it is not necessary to re-calculate the PI relative to the edge of each branch, which will not only be computationally expensive, but will also pose a limitation as around 50% of the cells are estimated to be located at branching points, and thus we would have to exclude them from the analysis.

Importantly, we showed that this pattern of polarity was very sensitive to manipulation in VEGF concentrations in the system. These observations strengthen the validity of the method and strongly supports its usefulness to measure endothelial collective behaviour *in vivo*.

**Author response image 1. respfig1:** Data supporting response to reviewers. (**A**) Analysis of polarity index and branch orientation in intraocular injections of PBS, VEGF, sFLT1 retinas. Left: examples of the sprouting front vascular network of designated treatments. Right: Box plots for Polarity Indexes, Branch Angle, and distribution of Mean Branch Angle and Mean Polarity Index for each retina. The analysis of branch direction was performed using Fiji. Briefly, the sprouting front of each retina was skeletonize and short branches were pruned. Subsequently, we used the directionality plugin to infer branch directionality. PI analysis and statistical analysis was performed in Matlab. n = 3 independent retinas. (**B**) Left: Example of area of analysis in a P6 mouse retina. Rectangles represent bins of 100µm wide used to calculate Polarity Index of endothelial cells towards the sprouting front (green > 0-100µm; blue > 300-400µm; yellow > 900-1000µm). Right: Distribution of Polarity Indexes along the Sprouting Front to Optic Nerve axis of the mouse retina, for retinas injected either with PBS (blue line), VEGFA (green line), sFLT1 (VEGFA inhibitor, red line), treated for 36h. Standard deviation is shown as lighter color. Note that close to the Sprouting Front, endothelial cells show a positive Polarity Index, meaning pointing towards the Sprouting Front, but closer to the Optic Nerve, endothelial cells show a negative value of Polarity Index, representing a polarity away from the Sprouting Front (towards the Optic Nerve). VEGFA treatment delays the transition from positive to negative values. sFLT1 reverts polarity of endothelial cells even at the Sprouting front, and in this condition, endothelial cells mainly polarize towards the Optic Nerve. This analysis was possible because of an adaptation of the Polarity Index formula, imposing a positive value to the Polarity Index if mean value of the resultant vector is between 0-179 degrees, and a negative value if mean value of the resultant vector is between 180-359 degrees. (**C**) Polarity index box plots of siWnt5a transduced or non-transduced cells with Vinculin-Full-Length-GFP (n = 6 independent experiments), Vinculin-T12-GFP (n = 6 independent experiments) or ɑCat-Vinc-HA GFP (n = 7 independent experiments). No significant differences were observed between transduced or non-transduced cells coming from the same experiment. (**D**) Quantification of mRNA expression levels of Wnt5a by qPCR showing the knockdown efficiencies of each single siRNA against Wnt5a (n = 6 replicas, from 3 independent experiments). (**E**) Western blot showing poor specificity of Wnt5a (~45 kDa) antibodies.

2) In some cell biology experiments, explanation of experimental protocols and data presentation seem insufficient in the text including sections of Materials and methods, Results and figure legends, which sometimes caused the reviewer difficulties in understanding the data completely. Specifically, for rescue or overexpression experiments using Lenti virus vector in scratch wound assay (Figure 1H and J, Figure 7 and Figure 8), did the authors use only transduced cells after cell sorting or not? If not, what percentage of cells used were transduced in each experiment? Transduction efficiency can potentially affect the experimental result. If they assayed in mosaic condition mixed with no transduced cells, they may wish to compare the effect between transduced and non-transduced cells in same dish.

We realised from this comment that our experimental design was not sufficiently clear in the Materials and methods section. We have updated the description with the hope that it will be clearer for readers.

In summary, we did not positively select for transduced cells during the direction of the experiment. Our analysis was performed using mixed populations. We have optimised viral protocols to achieve high transduction. In general, we have transduction efficiencies ranging from 35-75%. An extra step of enrichment was performed during imaging. Only regions with high-density of transduced cells was selected for analysis. We have nevertheless re-analysed the transduced vs. non-transduced cells. In this analysis, we obtained very similar values for PI for both transduced and non-transduced cells. We provide results in Author response image 1. We interpret this as a collective behaviour effect and likely means that tissue-level stresses are increased overall in the monolayer in rescue conditions, which are sufficient to sensitise and coordinate non-transduced Wnt5a KD cells. Yet, as this is very speculative we opted to not include this analysis in the manuscript, but to explain in more detail the way the experiment was performed in Materials and methods.

Also, biochemical data showing the levels of exogenous proteins expressed using overexpression vector were omitted in all rescue experiments.

Following the reviewer suggestion, we have now included WB data showing the efficiency and specificity in experiments in Figure 7—figure supplement 2 and Figure 8—figure supplement 1.

Specific points:1) The reviewer could not understand how "the correlation length" was calculated even though the reviewer referred to Ng et al., 2012 and Petitjean et al., 2010. More detailed information needs to be described anywhere in the text.

We added this information in Materials and methods, on a sub-section “Particle image velocimetry (PIV) analysis”.

2) Do the authors have any explanations as to why the coordination of leader cells is significantly affected by Wnt5A KD?

We thank the reviewer for this question. It is not an easy question to answer, as collective motility is influenced by several interdependent parameters. To clarify this question, we would like to refer the reviewer to Vitorino et al., 2008. In this elegant paper, the authors described in detail that endothelial collective cell migration depends on 3 semi-independent parameters, namely cell motility, directed behaviour, and coordination.

Endothelial cells are epithelial in nature, and have strong adhesive proprieties. For this reason, endothelial cells rarely detach from each other. Thus, leader cells remain attached to follower cells, and leaders are also influenced by the behaviour of followers. Moreover, we need to emphasise that, both *in vitro* and *in vivo*, endothelial leader cell is not a definitive position, but rather a dynamic state, in which leader cells can be replaced by follower cells, and vice-versa.

Based on Vitorino et al., along with many other reports in the field, we can postulate that the main reason for the impaired polarity of siWNT5a leader cells compared to siControl cells is due to a decrease in coordination of cell movements between leader and follower cells.

3) Biochemical data and the quantification shown in Figure 4A and B are confusing to the reviewer. β-catenin levels seem higher in control group than in Wit5A KD group in the representative data of western blotting in Figure 4A, while the mean level rather lower in the control in the quantification data of Figure 4B although there is no statistical significance. Which is correct? Also, vinculin levels are significantly higher in Wint5a KD group in Figure 4B, but they describe that there were not any differences there in the text (subsection “Non-canonical Wnt signaling regulates vinculin stability at adherens junctions to reinforce junctional mechanocoupling”).

We thank the reviewer for the thorough checking of the data. This pointed to a specific issue in the figure that is now corrected. The mistake was an interchange of labels of the bar plots from older version of the manuscript. Specifically, the former order from left to right was VE-cadherin, vinculin, a-catenin, b-catenin. Whilst the correct order was VE-cadherin, α-catenin, β-catenin, vinculin, which we have now corrected. We apologise for the oversight, and thank again the reviewer for this important correction.

Yet, this does not fully address the concerns raised by the reviewer.

a) Question1: β-catenin levels in blot and graph seem not to correspond. The WB blot image that we showed comes from one experiment of the 4 replicas. We showed the WB for all adherens junction proteins coming from the same protein extract, which might not reflect all the replicas. The experiment was repeated 4 times (during revision, we have performed one additional experiment to re-confirm everything). The result remains constant, with no statistical differences in β-catenin levels between siControl and siWnt5a cells. To clarify, we have now replaced the original WB data with blots from another biological replicate.

b) Question2: Regarding vinculin levels, they are not significantly upregulated. As mentioned before, the mistake came from an interchange of labels of the bar plots from older version of the manuscript. We have now corrected and added extra-quantifications. The data shows that there are no significant changes in vinculin protein levels.

4) Is the overall wound closure impaired in vinculin KD ECs concomitant with the decrease of PI?

We have quantified wound closure in vinculin KD cells, which shows a significant impairment. We have added the quantification data to Figure 6C.

5) The authors should change the description of Figure 8D to Figure 8E (subsection “Constitutively active vinculin is sufficient to rescue collective behavior defects in Wnt5a-deficient endothelial cells”).

We corrected the mistake.

6) Figure 9H needs to be quantified similarly.

We have quantified and present the results in Figure 9I.

7) There are no headings in each figure legend throughout the manuscript.

We have now included the headings to figure legends.

8) The authors need to state how they considered statistical significance in the text.

We are unsure about the exact meaning of the request. Nevertheless, believe that this might be related to define which condition is considered as statistically significant. We have added a sentence in the Materials and methods section stating that we consider significant a p value below 0.05.

9) The authors need to change the description to the correct one in the figure legend for Figure 1C.

We have corrected it.

Reviewer #3:[…] Overall the manuscript is well written, the data are comprehensive and generally support the conclusions. There are a few points that would strengthen it further.1) The claim for reduced association between actin stress fibers and VE-cadherin is not clear from the data shown (Figure 2B and Figure 9H). Some form of quantification is required to support this.

We have now quantified the number of actin stress fibers associated to adherens junctions in siControl, siWnt5a and siCdc42, which shows a significant decrease (Figure 2D, and Figure 9J).

2) Vinculin is shown to co-localize with VE-cadherin at junctions by immunofluorescence imaging (Figure 4C) yet by PLA, sites of co-localisation appears to be scattered across the cell at random and not enriched at junctions. What are the implications of this distribution pattern for how vinculin and VE-cadherin regulate junction tension?

We agree with the reviewer that our PLA experiments shows signals that are not only localized at junctions. However, this signal seems to be specific as no signal can be detected in control experiments (incubation with only one of the primary or no antibody). This is indeed a puzzling result. We can speculate about its meaning, yet we cannot provide a conclusive proof. These extra-junctional PLA signals can be derived from either pools of cadherin-catenin-vinculin complexes being recycled from junctions, establishment of ectopic cadherin-catenin-vinculin complexes at the cell membrane but not at junctions, or as suggested previously early cadherin-catenin complex assembly in the Golgi.

http://jcb.rupress.org/content/144/4/687

https://www.ncbi.nlm.nih.gov/pmc/articles/PMC2742162/

Nevertheless, as we were not completely sure about the source of those extra-junctional complexes, we only quantified junctional puncta, as described in the Materials and methods. We included additional information in the Materials and methods to provide clarity on the quantification.

3) Was the co-immunoprecipitation in Figure 4G performed on cells that had been wounded? If not, what is the source of tension that promoted VE-cad and vinculin co-association and would this be stronger in wounded monolayers?

Co-IP was done in wounded monolayers, as described in the Materials and methods. We added text to the Results section to clarify this point.

4) Retina vessels in vinculin ECKO mice (Figure 6C and D) show some phenotypic overlap with Wnt5a/Wnt11 mutants. This includes reduced vessel density which in Wnt5a/Wnt11 mice is due to increased vessel regression. Do the Vincluin ECKO mice have increased vessel regression? The authors should quantify this and discuss the results in the context of the junction mechanocoupling mechanism they describe.

We quantified vessel regression and we added the results to Figure 6 and discuss this result in the Discussion chapter.